# Targeting Tumor Glycans for Cancer Therapy: Successes, Limitations, and Perspectives

**DOI:** 10.3390/cancers14030645

**Published:** 2022-01-27

**Authors:** Nora Berois, Alvaro Pittini, Eduardo Osinaga

**Affiliations:** 1Laboratorio de Glicobiología e Inmunología Tumoral, Institut Pasteur de Montevideo, Montevideo 11400, Uruguay; apittini@pasteur.edu.uy; 2Departamento de Inmunobiología, Facultad de Medicina, Universidad de la República, Montevideo 11800, Uruguay

**Keywords:** tumor glycans, cancer immunotherapy, cancer glycobiology, monoclonal antibodies, cancer vaccines, CAR-T cells

## Abstract

**Simple Summary:**

Aberrant glycosylation is a common feature of many cancers, and it plays crucial roles in tumor development and biology. Cancer progression can be regulated by several physiopathological processes controlled by glycosylation, such as cell–cell adhesion, cell–matrix interaction, epithelial-to-mesenchymal transition, tumor proliferation, invasion, and metastasis. Different mechanisms of aberrant glycosylation lead to the formation of tumor-associated carbohydrate antigens (TACAs), which are suitable for selective cancer targeting, as well as novel antitumor immunotherapy approaches. This review summarizes the strategies developed in cancer immunotherapy targeting TACAs, analyzing molecular and cellular mechanisms and state-of-the-art methods in clinical oncology.

**Abstract:**

Aberrant glycosylation is a hallmark of cancer and can lead to changes that influence tumor behavior. Glycans can serve as a source of novel clinical biomarker developments, providing a set of specific targets for therapeutic intervention. Different mechanisms of aberrant glycosylation lead to the formation of tumor-associated carbohydrate antigens (TACAs) suitable for selective cancer-targeting therapy. The best characterized TACAs are truncated O-glycans (Tn, TF, and sialyl-Tn antigens), gangliosides (GD2, GD3, GM2, GM3, fucosyl-GM1), globo-serie glycans (Globo-H, SSEA-3, SSEA-4), Lewis antigens, and polysialic acid. In this review, we analyze strategies for cancer immunotherapy targeting TACAs, including different antibody developments, the production of vaccines, and the generation of CAR-T cells. Some approaches have been approved for clinical use, such as anti-GD2 antibodies. Moreover, in terms of the antitumor mechanisms against different TACAs, we show results of selected clinical trials, considering the horizons that have opened up as a result of recent developments in technologies used for cancer control.

## 1. Introduction

As the immunotherapy of cancer is a rapidly evolving field, advances in understanding the basic principles regulating the immune response are crucial. Significant progress in antibody engineering, as well as in adoptive cellular therapies, has determined that immunotherapy is one of the pillars of cancer treatment [1]. Three strategies achieved significant progress in clinical oncology: (A) monoclonal antibody (mAb) production against cancer molecular targets [2]; (B) immune checkpoint inhibitor development [3]; and (C) CAR-T lymphocyte generation [4].

Glycosylation is a complex form of post-translational modification, affecting over 50% of cell proteins and constituting a key regulator of many eukaryotic processes [5]. Aberrant glycosylation is a common feature of many cancers and plays crucial roles during all steps of tumor development [6]. Tumorigenesis and cancer progression can be regulated by glycosylation [7,8], controlling several physiopathological processes, such as cell–cell adhesion, cell–matrix interaction, epithelial-to-mesenchymal transition, tumor proliferation, invasion, metastasis, and angiogenesis [9,10,11]. Glycans can serve as a source for the development of novel clinical biomarkers [6], providing a set of specific targets for therapeutic intervention [8,12]. Different mechanisms of aberrant glycosylation lead to the formation of TACAs suitable for selective tumor targeting therapy [13]. In parallel, TACAs can interact with antigen-presenting cells through their interaction with glycan-binding receptors (such as SIGLECs, MGL, DC-SIGN), thus inducing immunosuppressive signals [14]. Based on these observations, several strategies using glycan-modified nanoparticles have been developed to improve antitumor immune responses [15]. Deeper knowledge of the structural and functional features of TACAs drives rational drug design, as well as novel antitumor immunotherapy strategies.

Aberrant glycosylation in cancer may occur in both glycoproteins and glycolipids. Among the best-characterized TACAs, we found truncated O-glycans (Thomsen-nouveau, Tn; Thomsen-Friedenreich, TF; and sialyl-Tn, STn), gangliosides (GD2, GD3, GM2, GM3, fucosyl-GM1), Globo-serie glycans (Globo-H, SSEA-3, SSEA-4), Lewis antigens, and polysialic acid. For more than twenty years, these TACAs have demonstrated potential usefulness in strategies for cancer immunotherapy [12,16]. In an attempt to classify and select tumor-associated antigens, the National Cancer Institute of the United States performed a comparative analysis following predefined objective criteria according to potential therapeutic effect, expression level, immunogenicity, and percentage of positive cells [17]. A listing of 75 tumor-associated antigens was established, and 9 of them were TACAs. The disialoganglioside GD2 was the best positioned TACA, in 12th place. The rest of the TACAs in the list are as follows: three other gangliosides (GD3, fucosyl-GM1, and GM3), Globo H, polysialic acid, as well as Tn, sialyl Tn, and sialyl-Lewis A (Table 1).

Therapeutic approaches involving TACAs have been diverse and include active immunity induced by vaccines, as well as monoclonal antibodies developed by genetic engineering. More recently, technologically advanced strategies, including bi-specific antibodies and chimeric antigen receptor-T (CAR-T) cells for anti-TACA, have been evaluated. However, as most TACA are poorly immunogenic, inducing of T cells independent immune responses, they must be conjugated to carrier proteins or be chemically modified to induce an effective antitumor immune response [18]. Much research has been conducted for anticancer vaccine development involving TACAs, and four of these studies achieved phase III clinical trials: Theratope^®^ (sTn), GM2-KLH, OPT-822 (Globo H) and Racotumomab^®^ [18]. On the other hand, passive antitumor immunotherapies consist of unconjugated antibodies displaying intrinsic cytolytic activity (antibody-dependent cell-mediated cytotoxicity, ADCC; antibody-dependent cellular phagocytosis, ADCP; complement-dependent cytotoxicity, CDC), antibodies conjugated to radionuclides (radioimmunotherapy), and antibodies conjugated to toxins (immunotoxins) or drugs (ADC). Some examples include antibodies specific to glycolipids (such as GM2, Globo H, and LeY), which can mediate cancer cell killing and tissue destruction by CDC [19], and antibodies recognizing aberrant expression of the O-glycosylated Tn antigen on the surface of cancer cells, inducing ADCC [20]. The use of Dinutuximab, a chimeric anti-GD2 antibody (ch14.18), for high-risk neuroblastoma (NB) treatment is a milestone, being the first new agent targeting a TACA approved by the U.S. Food and Drug Administration (FDA) in 2015 [21]. This review will focus on tumor glycans as targets for cancer therapy using different immunological approaches, considering their stage of progress in clinical trials.

## 2. Glycosphingolipids

Glycosphingolipids (GSLs) comprise a heterogeneous group of membrane lipids formed by a ceramide backbone covalently linked to a glycan moiety by a β-glycosidic bond [22]. Over 300 different GSLs are synthesized by enzymes localized in the ER and Golgi apparatus, integrating signaling components, assembly of glycosylating machinery and GSL trafficking, although not completely understood to date [23]. GSLs are ubiquitous components of cell membranes and are particularly abundant on the surfaces of animal cells, where they play an important role in molecular signaling, cellular adhesion, and motility [24]. In vertebrate animal tissues, they are subdivided into three major series: globo-, ganglio-, and neolactoseries [22].

The main GLSs characterized as TACAs are Globo H [25], the stage-specific embryonic antigens-3 and -4 (SSEA-3 and SSEA-4) [26], and the glycosphingolipids containing sialic acid such as the gangliosides GD2, GD3, GM2, fucosyl GM1, and Neu5GcGM3 [7]. These GSLs can affect cancer development by controlling cell adhesion, motility, and growth [27], the epithelial–mesenchymal transition and metastatic development [28], as well as drug resistance [29].

### 2.1. GD2

Thus far, the most relevant TACA concerning its clinical utility in antitumor immunotherapy is the disialylganglioside GD2. It is synthetized uniformly and abundantly in most neuroblastomas, melanomas, and retinoblastomas, as well as in several Ewing’s sarcomas [30], but it is not expressed in normal tissues, except for a weak expression in the brain, peripheral sensory fibers, and skin melanocytes [31]. GD2 promotes tumor cell proliferation, motility, migration, adhesion, invasivity, and confers apoptosis resistance, depending on cancer type [32]. Thus, anti-GD2 mAbs are able to directly induce cellular death without the involvement of immune mechanisms, thus combining apoptosis and necrosis features in tumor cell lines expressing GD2 [33,34]. GD2 is important for cancer cell survival and can suppress T cell activation and dendritic cell maturation when released into circulation [35].

Several anti-GD2 mAbs have been successfully tested in clinical practice, including chimeric 14.18 (ch 14.18) (NCT00026312) [36], humanized 14.18 (hu 14.18) [37], and humanized 3F8 (hu3F8) [38,39]. Passive immunotherapy with the anti-GD2 antibody Dinutuximab (ch14.18) became the first approved immunotherapeutic drug targeting a glycan antigen. Dinutuximab (Unituxin™, United Therapeutics Corporation, Silver Spring, MD, USA) is an IgG1 human/mouse chimeric Ab produced in mouse myeloma cell line SP2/0 [40], approved by the Food and Drug Administration (FDA) in 2015 for high-risk pediatric neuroblastoma treatment in combination with granulocyte-macrophage colony-stimulating factor (GM-CSF), interleukin-2 (IL-2), and 13-cis retinoic acid (RA) [21]. Subsequently, Dinutuximab beta (ch14.18.CHO; Qarziba™, EUSA Pharma, Hemel Hempstead, UK), produced in Chinese hamster ovary (CHO) cells, was approved by the European Commission in 2017 for treatment in high-risk neuroblastoma patients. Both antibodies are currently part of the standard-of-care of neuroblastoma. Regarding immunological mechanisms, Dinutuximab binds to GD2 and induces ADCC as well as CDC and subsequent neuroblastoma cell death by recruiting granulocytes and natural killer cells from peripheral blood mononuclear populations [41,42]. More recently, the FDA granted accelerated approval to humanized mAb Hu3F8 (Naxitamab, DANYELZA^®^) in combination with GM-CSF for relapsed or refractory high-risk neuroblastoma in the bone or bone marrow, based on overall response rate and duration of response in two clinical trials (NCT03363373 and NCT01757626). Recently, another clinical trial (NCT04560166) began evaluating Naxitamab and GM-CSF in combination with irinotecan and temozolomide for patients with primary refractory high-risk NB or in their first relapse. Table 2 shows selected clinical trials of antibodies targeting TACAs.

In melanoma, the antitumor activity of the hu14.18-IL2 immunocytokine was evaluated in a minimal residual disease setting (NCT00590824), and prolonged tumor-free survival was seen in some treated melanoma patients at risk of recurrence [46]. Toxicity of the radiolabeled 131I-mAb 3F8 intrathecal was first evaluated in refractory neuroblastoma (NCT00450827), and a phase II trial is ongoing to evaluate the radiolabeled antibody for the treatment of central nervous system cancer or leptomeningeal metastasis of several tumors expressing GD2 (NCT00445965).

Currently, several clinical trials are also assessing anti-GD2 treatments in other tumors expressing disialylganglioside. In osteosarcoma, Dinutuximab combined with GM-CSF in recurrent patients to characterize the pharmacokinetics and toxicity (NCT02484443), and Hu3F8 along with GM-CSF in patients in remission, evaluating its effectiveness in preventing recurrence (NCT02502786). In addition to unlabelled anti-GD2 mAbs, other therapeutic approaches include the development of immunocytokines, immunotoxins, antibody–drug conjugates, radiolabeled antibodies, targeted nanoparticles, vaccines, CAR-T cells, as well as bispecific antibodies (BsAbs, antibodies with two binding sites directed at two different antigens [48,60]).

BsAbs targeting GD2 offer great promise in anticancer therapy [61]. Some of these BsAbs combining anti-GD2 with anti-immune cell receptors are the anti-GD2 mAb 7A4/anti-Fc gamma RI (CD64) [62], the murine 5F11-scFv/anti-CD3 [63], the humanized hu3F8-scFv/anti-CD3 [62], and hu3F8/anti-CD3 [64]. This strategy brings the immune cells into close proximity with cancer cells and leads to killing them through perforin/granzyme-mediated, non-MHC-restricted, specific antitumor cytotoxicity. Some of these BsAbs have also shown a powerful antitumor response in neuroblastoma models [62,65] and are currently being evaluated in clinical trials. A phase I/II trial is assessing the toxicity, pharmacokinetics, and antitumor activity of hu3F8-CD3 in relapsed-refractory neuroblastoma, osteosarcoma, and other GD2-expressing solid tumors (NCT03860207), while another multicentric trial is evaluating it in small cell lung cancer (SCLC) (NCT04750239). Recently, a novel strategy using a bispecific-trifunctional antibody (trABs) combining an IgG molecule of both anti-GD2 and anti-CD3 specificity, as well as the innate immunity activation via their functioning Fc-fragment, was able to induce a better antitumor response compared with mAb ch14.18 [66].

GD2, being a “self” antigen, is a poor immunogen and it is difficult to induce a specific anti-GD2 immune response in vivo. With the aim of improving its immunogenicity, GD2 conjugated to the keyhole limpet hemocyanin (KLH) as a carrier protein was assayed with QS-21 or OPT-821 as adjuvants. A bivalent vaccine, containing GD2- and GD3- linked to KLH, with OPT-821 as adjuvant, was first evaluated in high-risk neuroblastoma patients in a phase I trial designed to find the maximally tolerated dose and antibody responses against GD2 and/or GD3. Such a response was observed in 12 of 15 patients, and the disappearance of minimal residual disease (MRD) was documented in 6 of 10 patients [67]; moreover, a larger, phase II trial (NCT00911560) is currently demonstrating robust antibody responses, associating higher anti-GD2-IgG1 titer with improved survival [68]. A trivalent vaccine, containing GM2-GD2- and GD3- also linked to KLH, with OPT-821, was evaluated in patients with metastatic sarcoma, inducing mostly the IgM response, without clinical benefit (NCT01141491). Table 3 shows selected clinical trials of antitumor vaccines based on TACAs.

### 2.2. GD3

GD3 is another di-sialic ganglioside only synthetized during development and in pathological conditions such as neurodegenerative illnesses [81], as well as in cancers of a neuroectodermal origin [82]. It is also detected in neural stem cells in which it interacts with EGFR to sustain EGF-induced downstream signaling to maintain the self-renewal capability of these cells [83]. ST8SIA1, also known as GD3 synthase (GD3S), is the only enzyme that regulates the biosynthesis of GD3 and GD2. It has been demonstrated that overexpression of GD3 and GD3S in glioblastoma stem cells plays a key role in tumorigenesis by expression of stemness genes and self-renewal potential [84] and the enzyme is frequently overexpressed in other tumors such as breast cancer, melanoma, lung cancer, and hepatocarcinoma, leading to GD3S being proposed as novel drug targets in cancer [85]. GD3 was reported to be a melanoma-associated ganglioside, detected in the majority of human melanoma tissues and cell lines, but not in normal melanocytes [86]. While GD2 and GD3 enhanced cell growth, GD3 might also contribute to cell invasion. Using a real-time adhesion assay in a melanoma cell model, Ohkawa et al. demonstrated that GD3 positive cells exhibited stronger adhesion properties than controls in an extracellular matrix, especially to collagen type I and type IV [87]. Upregulation of GD3 has been associated with melanoma progression, brain metastases, and poor outcomes [88].

GD3 is considered a marker of neuroectoderm origin in tumors, and several mAbs have been developed from mice immunized with the human melanoma cell line SK-MEL-28, e.g., R24 [89]; 2B2, IF4 and MG-21 [90]; K641 [91]. MAb R24 was further characterized as anti-GD3 with reactivity restricted to melanocytes, neuronal and glial cells in the central nervous system, parotid and adrenal cells [92], and it exhibited activation of complement and ADCC in preclinical models [93]. Multiple clinical trials have been conducted using R24 alone or in combination with other drugs, exhibiting variable rates in clinical responses; however, a significant concern was human anti-mouse antibodies’ (HAMA) production [93]. A mouse–human chimeric R24 (chR24) molecule was constructed, but chR24 demonstrated a lower level of binding to GD3 than mouse R24 [94]. In the last three years, a clinical trial phase I for a novel drug conjugated monoclonal antibody (PF-06688992), composed of humanized anti-GD3 huR24 linked to a chemotherapeutic agent, was conducted in stage III-IV melanoma patients (NCT03159117). The purpose of the study was to evaluate the safety and efficacy of this drug that had never before been given to people; results are still forthcoming. Another monoclonal antibody characterized by a high affinity for GD3 is KM641 [91], found to cause in vitro cytotoxicity by both CDC and ADCC. The chimeric version of this antibody (KW2871, ecromeximab) demonstrated, in preclinical and phase I studies, biodistribution, long half-life, and lack of immunogenicity in patients with metastatic melanoma [95]. Other clinical trial confirmed safety and established the maximum tolerated dose of 40 mg/m^2^ (NCT00199342) [49]. However, a subsequent phase II study in metastatic melanoma conducted by Tarhini et al., combining ecromeximab with high dose of interferon-α2, looking for progression-free survival and response rate, concluded that, although generally well tolerated and with low immunogenicity, the clinical benefit was limited in this combination, suggesting the need to evaluate other strategies (NCT00679289) [50]

Considering vaccine strategies, because GD3 is poorly immunogenic, Chapman and Houghton developed anti-idiotypic mAbs against the well-characterized anti-GD3 mAb R24 and confirmed that BEC2 could mimic GD3 gangliosides and induce anti-GD3 IgG in rabbits [96]. BEC2 was evaluated in melanoma patients, comparing BCG and QS21 as adjuvants [97], and as BEC2/BCG gave better results; thus, it was evaluated further in other cancers. In 1999, Grant et al. evaluated BEC2/BCG as an adjuvant therapy in patients with SCLC who had completed standard therapy and achieved substantially better survival than those observed in a prior group of similar patients (NCT 00037713) [69]. However, two later phase III studies in SCLC patients with limited disease, receiving five vaccinations of BEC2 (2.5mg)/BCG vaccine, failed to demonstrate improvements in survival, progression-free survival, or quality of life in the treated group (NCT00006352; NCT00003279) [70,71].

### 2.3. Fucosyl-GM1

The ganglioside fucosyl-GM1 (FucGM1) is a tumor-associated antigen strongly detected in a large percentage of human SCLC but absent in most normal adult tissues [98,99]. The restricted normal tissue detection and the overexpression on SCLC suggest that FucGM1 could be an attractive target for both active and passive immunotherapy against this highly aggressive cancer [100]. Vaccination with synthetic fucosyl-GM1-KLH conjugate induces an antibody response against cancer cells expressing fucosyl GM1 and CDC against SCLC cells [101,102]. However, the antibody response is comprised predominantly of low-affinity IgM antibodies, while the IgG response to this vaccine was low (only one of six patients at the highest dose had an IgG antibody titer of 1:80 or greater) [102].

The potential therapeutic effect of the mAbs anti-FucGM1 was established in vivo for the first time in experimental models, demonstrating growth inhibition of SCLC tumors [103] and a rat hepatoma tumor [104]. More recently, a novel non-fucosylated fully human IgG1 antibody (BMS-986012), specific to FucGM1, was developed [105]. BMS-986012 showed high binding affinity for FcγRIIIa (CD16), resulting in enhanced ADCC, as well as CDC and ADCP against FucGM1-expressing tumor cell lines [105]. This mAb induced tumoral regression in a SCLC xenograft model (DMS79). Antitumor activity was enhanced when BMS-986012 was combined with cisplatin or etoposide standard-of-care therapy. These preclinical data support the evaluation of BMS-986012 in a phase I/II clinical trial as first-line therapy in patients with extensive-stage SCLC (NCT02815592). Initial results of another phase I/II study evaluating BMS-986012 alone and in combination with nivolumab in patients with relapsed/refractory SCLC (NCT02247349) demonstrate that this mAb is well tolerated and shows evidence of antitumor activity in some patients [51]. Finally, an ongoing phase II study is evaluating whether BMS-986012 in combination with carboplatin, etoposide, and nivolumab is able to improve overall survival in newly diagnosed extensive-stage SCLC patients compared with carboplatin, etoposide, and nivolumab alone (NCT04702880).

### 2.4. GM3

Ganglioside GM3 is widely distributed in essentially all types of animal cells, and overexpressed in several types of human cancers, such as melanomas, lung, and brain cancers [106,107]. Its detection at the surface of tumor cells is an important factor in determining the metastatic phenotype [108,109]. It has been demonstrated that GM3 density may affect antigenicity. The mAb M2590 reacts with melanoma cells, but not with corresponding normal tissue [110]; it was unexpectedly proven that the identified epitope was GM3. The recognition of this molecule does not occur in a monomeric state, and it was shown that a high ganglioside density was required for the reactivity of M2590 mAb, suggesting that the cancer specificity displayed by this antibody is established by the recognition of a more densely packed cluster state of GM3 [111].

Several variants of GM3 are relevant in oncology. The de-N-acetyl GM3 (d-GM3), a variant exhibiting a free amino group at position 5 of the sialic acid instead of the acetyl group, was found in melanoma, and it was demonstrated that it enhances cell migration and invasion [112]. A great interest in antitumor immunotherapy was demonstrated for the variant GM3, which contains N-glycolylneuraminic acid (Neu5Gc) instead of N-acetylneuraminic acid (Neu5Ac) [113]. GM3(Neu5Gc) is widely synthetized in most mammals, but not in normal human cells, due to an inactivating mutation in the human cytidine monophospho-N-acetylneuraminic acid hydroxylase gene [114]. In contrast, GM3(Neu5Gc) is often highly detected in human cancers, such as melanoma, colon, breast, and lung cancer [115,116,117,118], a phenomenon that has been attributed to Neu5Gc metabolic incorporation into cancer cells from dietary sources, particularly red meat [116]. The ganglioside GM3 (Neu5Gc) is a neoantigen with promising possibilities for cancer immunotherapy [119]. Several GM3 (Neu5Gc)-targeting antibodies have been developed, and the best characterized are the mouse IgG1 mAb14F7 [120] and its humanized variant 14F7hT [121]. Additionally, 14F7hT shows potent in vivo antitumor ADCC on a solid mouse myeloma model [122], as well as on SKOV3 human ovarian carcinoma cell and Lewis lung carcinoma (3LL) mouse tumors [123].

Considering that gangliosides display low immunogenicity, a strategy for vaccine development was the solubilization of the hydrophobic outer membrane proteins of *Neisseria meningitidis* with GM3 (Neu5Gc) to form small-sized proteoliposomes (VSSP) (GlycoVaxGM3-NeuGcGM3/VSSP) [124]. This vaccine was reported as safe and immunogenic [119]. Vaccination of melanoma patients [125] and breast cancer patients [126,127] induced high titers of antibodies IgM and IgG anti-GM3 (Neu5Gc). In melanoma patients, the GlycoVaxGM3 vaccine improved overall survival of metastatic patients after first-line chemotherapy [128]. GlycoVaxGM3 clinical effectiveness was also evaluated in metastatic breast cancer patients, and overall survival was higher in vaccinated patients [129].

Another strategy to induce an immune response by anti-TACA is the use of peptide structures simulating the tumor glycan, such as immunization with anti-idiotypic antibodies, following the theory of idiotypic network proposed by Jerne [130]. This strategy is feasible as a means by which to induce an immune response against some TACAs, such as GD2 [131] and GD3 [97]. Although several preclinical studies using anti-idiotypic antibodies supported their utility as antitumor vaccines, human studies have been disappointing and anti-idiotypic vaccines failed in clinical trials [132]. However, promising results were obtained with the anti-idiotype vaccine Racotumomab (Vaxira^®^), which mimics the ganglioside GM3 (Neu5Gc). The first step in Racotumomab development was mice immunization with liposomes containing GM3 (Neu5Gc), and the result was the mAb P3 (IgM), which recognized different cancers expressing GM3 (Neu5Gc). Subsequently, mice immunization with the mAb P3 conjugated to the carrier protein KLH led to the obtaining of anti-idiotype mAb 1E10 (IgG) (Racotumomab) [133]. A phase I clinical trial of Racotumomab was conducted in children with a diagnosis of cancers expressing N-glycolylated gangliosides, resistant to conventional therapy (NCT01598454), confirming a favorable toxicity profile up to a dose of 0.4 mg, and most patients elicited an immune response [72]. Currently, a phase II trial is ongoing in patients with high-risk neuroblastoma (NCT02998983). Two clinical trials evaluated this mAb in advanced lung cancer (NCT01240447 and NCT01460472). Vaxira^®^ has reached the market and is the first approved anti-idiotype vaccine (in Argentina and Cuba) as an active immunotherapy agent for advanced non-small cell lung cancer (NSCLC) treatment. It was found that anti-GM3 (Neu5Gc) Abs induced by Racotumomab vaccination can mediate an antigen-specific ADCC response against tumor cells in NSCLC patients [134]. In the same way, an immunological response producing IFNγ was found in metastatic breast cancer patients treated with Vaxira^®^ [135].

### 2.5. Globo-Series

GSLs of the globo-series, such as stage-specific embryonic antigen 3 (SSEA-3), SSEA-4, and Globo-H, are specifically synthetized on pluripotent stem cells and cancer cells. They are known to be associated with various biological processes such as cell recognition, cell adhesion, and signal transduction [26,136]. Among them, the most prevalent cancer-associated antigen is Globo-H ceramide (GHCer), which is overexpressed in several cancers, including breast, gastric, lung, ovarian, endometrial, pancreatic, and prostate cancers [99,137]. The detection of GHCer in normal tissues is restricted to the luminal surface of glandular tissues, usually not accessible to the immune system [25]. GHCer could be transferred from cancer cells into non-tumor cells located in tumor microenvironment, such as endothelial cells, promoting angiogenesis [138], as well as into T cells, inhibiting IL-2, interferon-γ, and IL-4 secretion, promoting immunosuppression [25]. The specific expression of GHCer in tumor cells, as well as its role promoting tumor progression, makes this antigen an attractive target for anticancer immunotherapies.

GHCer-targeted immunotherapy in breast cancer has generated encouraging results, and a vaccine containing Globo H–KLH conjugate plus the immunological adjuvant QS-21 has been shown to be safe, along with induction of humoral antibody responses, in two phase I clinical trials carried out in patients with relapsed prostate cancer [139] and metastatic breast cancer [140]. More recently, a multi-national randomized phase II trial of Globo H–KLH with adjuvant OBI-821 vaccine was evaluated in 348 patients with metastatic breast cancer (NCT01516307). The authors found that, although there was no difference in progression-free survival between patients treated with Globo H vaccine and those treated with placebo, in the vaccinated group, those patients who mounted anti-Globo H responses had significantly better progression-free survival than the placebo group [73]. Based on these promising results, a global phase III trial in triple-negative breast cancer is ongoing (NCT03562637).

Regarding passive immunotherapy, tumor growth inhibition (TGI) was found in mice xenografted with human breast cancer MCF7 cells by treatment with anti-Globo-H (mAb VK9) and anti-SSEA4 (mAb MC813-70), showing 45% and 24% TGI, respectively, and 56% of TGI in combination treatment; this indicated that SSEA4 and Globo-H may play a synergistic role in regulating tumor growth [141]. Ruggiero et al. demonstrated that mAb MC-813-70 is rapidly internalized into triple-negative breast cancer cells following its binding to a specific target at the plasma membrane, accumulating in acidic organelles [142]. As such, conjugating this antibody with the saporin toxin, these authors developed an immunotoxin able to reduce the viability of breast cancer cells in vitro and in vivo. Furthermore, a novel antibody–drug conjugate (OBI-999), derived from an anti-Globo-H mAb conjugated with a monomethyl auristatin E (MMAE), displays excellent tumor inhibition in different animal models, including breast, gastric, pancreatic, and lung cancers [143]. Further work is needed to validate these novel drugs for the treatment of solid tumors; hence, two phase I/II clinical trials are currently evaluating the safety, pharmacocynetics, and therapeutic activity of the anti-globo H mAb OBI-888 in multiple advanced and metastatic solid tumors (NCT03573544), as well as the immunotoxin OBI-999 in advanced solid tumors (NCT04084366).

## 3. Simple Mucin-Type O-Glycan Antigens

Simple mucin-type O-glycosylated TACAs, such as the Tn antigen (CD175), the TF antigen (or T antigen, CD176), and the sialyl-Tn antigen (STn, CD175s), are attractive targets for anticancer therapies because they are detected in most carcinomas and are usually absent in healthy tissues [144]. Mucin-type GalNAc O-glycans are built up by a sequential step-by-step process in the Golgi apparatus, starting with the addition of a N-acetyl-galactosamine (GalNAc) to a serine or a threonine residue, whose result is the core GalNAcα-O-Ser/Thr (Tn antigen). This initial key step of the O-glycosylation process is catalyzed by the UDP-GalNAc:polypeptide-N-acetyl-galactosaminyl-transferases family (GalNAc-T) [145]. GalNAcα-O-Ser/Thr is then further elongated by other glycosyltransferases to generate complex O-glycans. For example, the Tn structure is the acceptor substrate for core 1 β3-galactosyltransferase (C1GalT1) to generate the core 1 disaccharide O-glycan Galβ1-3GalNAcα-O-Ser/Thr, (TF antigen). Moreover, addition of sialic acid to the Tn antigen, catalyzed by the ST6GalNAc-I, results in the sialyl-Tn antigen synthesis. The sialylation of the Tn antigen blocks further elongation of the saccharide chain. These truncated O-glycan antigens are observed at the earliest stages of cellular malignant transformation [146,147,148] and are significantly associated with tumor progression through various mechanisms affecting adhesion properties of cancer cells, stabilizing receptor expression on the cell surface allowing stronger signaling, and triggering immune suppression by binding to tolerogenic dendritic cells or modulating NK cells action by competitive lectin binding [149,150,151].

### 3.1. Tn Antigen

Different mechanisms can cause the cumulative expression of the Tn antigen found in cancer cells. Defects in the chaperone Cosmc due to epigenetic silencing or mutations, essential for the activity of the C1GalT1 enzyme, [152,153], but also changes in the expression and organization of different glycosyltransferases, as well as relocation of GalNAc-transferases [154,155], can play a role. Several mAbs recognizing the Tn antigen have been generated using different immunization strategies. Some of these antibodies are CU-1 [156], Ca3638 [157], MLS128 [158], BRIC 111 [159], 5F4 [160], HB-Tn1 [161], PMH1 [162], 83D4 [163], SM3 [164], 237mAb [165], PankoMab [166], 5E5 [167], KM3413 [168], 2154F12A4 [169], GOD3-2C4 [170], Kt-IgM-8 [171], 6C5 [172], and Remab6 [173]. Although the chemical structure of the Tn determinant is known to be GalNAcα-O-Ser/Thr, its immunological definition is more complicated, and some antibodies clearly require more complex epitopes than a single Tn residue. Indeed, some anti-Tn antibodies require the involvement of additional amino acids in the antigenic determinant. This is the case for the mAb FDC-6, which reacts specifically with a Tn on a hexapeptide sequence (VTHPGY) at fibronectin [174]; PMH1 can react with single or multiple Tn on a specific MUC2 apomucin peptidic chain [162]; mAb 237 is directed to a Tn-glycopeptide in murine podoplanin [165]. Several mAbs such as B27.29 [175], SM3 [164], PankoMab [166], and 5E5 [167] specifically recognize the Tn-MUC1 peptide, while mAb 6C5 is specific for a Tn-peptide epitope in dysadherin/FXYD5 [172]. These observations indicate that the neighboring peptide backbone could be an important factor that modulates the structure of the Tn epitope. In addition, the mAb PODO447 reacts with an unusual terminal motif (N-acetylgalactosamine beta-1, GalNAcβ1), predominantly found on the Podocalyxin (Podxl) protein core [176], a glycomotif rarely found in normal eukaryotic cells. Although some proteins are able to recognize a single Tn determinant, such as mAb Kt-IgM-8 [171] and plant lectins (VVLB4 and *Salvia sclarea*) [163,177], other anti-Tn monoclonal antibodies require at least two consecutive Tn residues for binding, such as MLS128 [178], 83D4 [163], KM3413 [168], and Remab6 [173]. This fact may be related to a higher specificity in cancer cell recognition. In addition, we demonstrate that the Tn backbone (serine or threonine) is important for some aspects of mAbs binding. Using synthetic Tn-based vaccines, we generated a panel of anti-Tn monoclonal antibodies able to recognize tri-Tn build on different backbones such as S*S*S*, S*T*T* or T*T*T* (S = Serine; T = Threonine; * = GalNAc). For example, the mAb 15G9 specifically recognized S*S*S* but failed to bind to the other structures [179]. These antibodies exhibited differentiated recognition by immunohistochemistry in human breast and colon cancer, demonstrating that the amino acid carrier of the GalNAc (Ser vs. Thr) could play a key role in anti-Tn specificity for cancer detection. This fact involves an additional complexity regarding the apparently simple structure of the Tn-antigen. The great variety of anti-Tn antibodies developed, exhibiting high differences in recognition patterns, demonstrates that, for several of them, the determinant GalNAcα-O-Ser/Thr is a necessary condition, but it may still be insufficient. In the same way, functional evaluation of different anti-Tn antibodies with arrays of synthetic saccharides, glycopeptides, and O-glycoproteins, demonstrates similar variability in fine specificity [180,181].

In vivo experiments showing a reduction in tumor growth mediated by anti-Tn antibodies can be explained by different mechanisms: (i) antibody-dependent cellular cytotoxicity (ADCC) [20,166,168,170,182,183]; (ii) antibody-induced complement-dependent cytotoxicity (CDC) [171]; (iii) inhibition of cancer cell adhesion to lymphatic endothelium [169]; and (iv) direct blocking of receptor signaling, such as epidermal growth factor receptor and insulin-like growth factor I receptor [184]. These different mechanisms may be strongly dependent not only of the antibody class but also of the fine specificity of each antibody.

With the aim of using anti-Tn antibodies as immunotherapeutic agents in cancer treatment, chimeric or humanized antibodies were generated from the variable region sequences of existing murine monoclonal antibodies: (i) cKM3413 is a mouse–human chimeric IgG1 antibody generated from the mAb KM3413 [168]. In vitro assays with Jurkat cells (a human T-lymphoid leukemia cell line) revealed that cKM3413 induced ADCC and direct killing activity with cross-link antibodies. In vivo experiments with Jurkat-inoculated C.B-17/lcr-scid Jcl mice showed significantly better survival in the group treated with cKM3413 compared with the PBS control group, suggesting the therapeutic potential of this antibody. (ii) Remab6 is a chimeric human IgG1 antibody [173] derived in part from the murine mAb Ca3638 [157]. Remab6 showed high specificity for cancer tissues. Its antitumor activity has not yet been reported. (iii) Natural MUC1 Abs from breast cancer patients reacted more strongly with GalNAc peptides than with the naked 60-mer peptide, indicating that a MUC1 glycopeptide could be a better vaccine [185]. The mAb 5E5 recognizes an immunodominant cancer-specific epitope, Tn-MUC1 (in the GSTA region), not covered by immunological tolerance in MUC1-humanized mice [167]. Moreover, 5E5 has been shown to lyse MCF-7 and T47D breast cancer cells via both ADCC and CDC [182]. Recently, it was found that humanized mAb 5E5 antibodies (CIM301-1 and CIM301-8) are potent enhancers of NK cell activation and cytotoxicity in vitro in MUC1-Tn/STn-positive tumor cells [183]. It was observed that, by removing the fucose on the N-glycan of the Fc tail (antibody CIM301-8), an increase in Fc-binding affinity to the FcγRIIIa was achieved, leading to enhanced ADCC by a potentiated NK cell response. (iv) PankoMab also recognized an immunodominant Tn-MUC1 determinant located in the PDTRP region [166]. This antibody showed high and specific ADCC-mediated lytic activity using human peripheral blood mononuclear cells (PBMC) as effector cells. PankoMab-GEX™ is a fully humanized antibody derived from PankoMab, also known as Gatipotuzumab, glycoengineered to achieve enhanced Fc-mediated antitumor activity [52]. A phase I clinical trial in patients with advanced solid tumors, mostly colon and ovarian cancers, demonstrated that the drug was safe and well tolerated (NCT01222624). Following promising preliminary efficacy in patients with ovarian cancer, a phase II was conducted in patients with recurrent ovarian, fallopian tube, or primary peritoneal cancer to evaluate the efficacy of PankoMab-GEX vs. placebo in maintaining response after chemotherapy (NCT01899599). However, this study failed to demonstrate an improvement in progression-free survival [53]. Another phase I trial in solid tumors evaluated the combination of Gatipotuzumab with an anti-EGFR antibody (NCT03360734), demonstrating feasibility and antitumor activity in colorectal cancer (CRC) and NSCLC patients [54]. (v) We generated and characterized the mAb Chi-Tn, a chimeric mouse/human antibody containing mAb 83D4 VH and VK variable regions and the constant region of IgG1κ human immunoglobulin [186]. The mAb 843D4 was generated from splenocytes of mice immunized with a human breast cancer [187] and required at least two consecutive Tn residues for antigen binding [163]. The binding of 83D4 is not influenced by the peptide structure [179]. The Chi-Tn antibody induced the rejection by ADCC of a murine breast cancer in 80% to 100% of immunocompetent mice, with this activity being strongly potentiated by cyclophosphamide [20]. The Chi-Tn rapidly internalizes into cancer cells and delivers cytotoxic drugs in active form, and the conjugated auristatin F exhibited efficient antitumor activity in vivo [188]. In addition, ChiTn enhanced internalization of nanoparticles containing docetaxel, inducing a strong reduction in the viability of human lung cancer cells [189].

Regarding the usefulness of the Tn antigen to induce immunological antitumor response, the first studies were performed by George Springer, using red blood cells expressing Tn/TF antigens. Immunization of advanced breast cancer patients with these antigens induced immune responses correlated with better survival, suggesting a novel strategy for cancer treatment [190]. However, the first synthetic Tn vaccine reported in clinical trials, a palmitoyl-Tn conjugate, failed to induce anti-Tn responses [191]. The fact that the binding of some anti-Tn monoclonal antibodies requires several consecutive Tn residues to recognize tumor cells is consistent with the fact that this antigen is known to be displayed as clusters on native mucins. This provided the basis for the rational design of glycopeptide-based anticancer vaccines containing glycotopes organized in clusters. Kuduk et al. [192] observed that clusters of three consecutive Tn residues, covalently linked to the carrier protein KLH, were able to induce high IgM and IgG anti-Tn antibody titers in mice. These antibodies were strongly reactive with the Tn positive human colon cancer cell line LSC, but not with the Tn-negative LSB cell line. A fully synthetic Tn immunogen that does not require a protein carrier, called multiple antigenic glycopeptide (MAG), was developed carrying the Tn antigen associated with a T-helper epitope (initially a poliovirus peptide). It was found that the MAG containing the tri-Tn glycotope was much more efficient than the mono-Tn analogue in promoting the survival of mice grafted with the mammary adenocarcinoma TA3/Ha cell line in immunotherapeutic experimental settings [193]. To ensure a broad coverage within the human population, the tetanus toxoid-derived peptide TT830-844 was selected as a T-helper epitope in MAG-Tn because it can bind to HLA-DRB molecules, largely expressed in the population [194]. This MAG-Tn-TT, in association with AS15, has been found to be well tolerated in non-human primates and induced robust Tn-specific IgM and IgG responses [194]. The MAG-Tn-TT vaccine was recently evaluated in seven patients with localized breast cancer with a high-risk of relapse. This phase I clinical trial (NCT02364492) demonstrated that all vaccinated patients developed high levels of Tn-specific antibodies that killed Tn-expressing human tumor cells through a CDC mechanism [74].

However, the Tn antigen, besides inducing antitumor immune response, can also favor tumor immune tolerance, explaining at least in part the failure of some vaccine strategies. For example, Tn is recognized by the tolerogenic lectin—macrophage galactose C-type lectin (MGL)—expressed by dendritic cells and macrophages, which allows these cells to suppress T cell immunity [195] and promote angiogenesis [196], thus playing a role in cancer progression. In addition, overexpression of Tn antigen in mouse MC38 colorectal cancer cells increased myeloid-derived suppressor cells and decreased CD8+ T cell infiltration, promoting an immune-suppressive tumor microenvironment [197].

### 3.2. Sialyl-Tn Antigen

STn is often coexpressed with Tn and, as Tn, its detection is low or null on normal cells or tissues, depending on mAb used. Aberrant STn expression is associated with dysregulation of the O-glycosylation machinery, including imbalanced expression of Cosmc and STn synthase (ST6GalNAc-I). De novo STn expression via ST6GalNAc-I transfection can change a tumor’s malignant phenotype [151], leading to more aggressive cancer cell behavior, decreasing cell–cell aggregation and increasing tumor growth, extracellular matrix adhesion, migration, invasion, and metastases [198,199,200]. In addition, it was demonstrated that STn-expressing cancer cells impair maturation of DCs, endowing a tolerogenic function and therefore limiting their capacity to trigger protective antitumor T cell responses [201].

The first evidence of STn detection in cancers was obtained by B72.3 mAb, produced by somatic fusion of splenocytes from mice immunized with human breast cancer membranes [202]. B72.3 is immunoreactive with a high molecular weight glycoprotein complex, designated tumor-associated glycoprotein TAG-72 [203]. The second-generation antibody CC49 was generated following mice immunization with TAG-72 molecules affinity purified by B72.3 [204]. Two other anti-STn mAbs, TKH2 [205] and HB-STn1, were generated by mice immunization with ovine submaxillary mucin (OSM). In the same way as anti-Tn antibodies, some anti-STn mAbs display better recognition when antigens are clusters of STn. For example, mAbs B72.3 and MLS 102 strongly bind glycoproteins bearing STn-trimers but exhibit poor interactions with monomeric-STn glycoproteins [206]. Similar results were observed by Ogata et al. for mAbs TKH2 and B72.3, both reacting with trimeric STn, but mAb TKH2 demonstrated greater binding than mAb B72.3 to monomeric STn [207]. Reddish et al. observed that mAb B72.3 chiefly recognized STn-serine clusters, but also exhibited cross-reactivity with the non-sialylated Tn-serine clusters [208]. By contrast, the mAb CC49 showed strong reactivity with STn-serine clusters and weak reactivity with Tn-serine clusters.

Compared to B72.3, mAb CC49 has an affinity constant about six times higher and has shown a 16-fold increase of tumor/blood ratio in human xenografts in athymic mice [209]. Murine CC49 and its humanized version (huCC49) [210] have been evaluated in phase I/II radioimmunotherapy clinical trials with encouraging results. Intraperitoneal radioimmunotherapy with 177Lu-CC49 was well tolerated and seems to have antitumor activity against chemotherapy-resistant ovarian cancer in the peritoneal cavity [211,212,213]. In a phase II clinical trial, radioimmunotherapy with 131I-CC49 in hormone-resistant metastatic prostate cancer, in combination with IFN, enhanced tumor uptake and antitumor effects compared to a prior phase II trial of 131I-CC49 alone [214]. Myelotoxicity is dose-limiting because of prolonged circulation time in the plasma, and HAMA responses were observed in most patients. To ameliorate these problems, a CH2 domain-deleted humanized CC49 (HuCC49ΔCh2) was developed [215]. Recently, in a murine model of ovarian cancer radioimmunotherapy using 225Ac-Labeled DOTAylated-huCC49 antibody, a significant reduction in tumor growth was observed in a dose-dependent manner, and survival was improved by more than three-fold compared with the untreated control group, without significant off-target toxicity [216]. Another therapeutic strategy with anti-STn antibodies is drug delivery. PEG-immunoliposomes (PILs) were prepared by conjugation of Fab’ fragments of huCC49 to target STn-overexpressing cancer cells [217]. These anti-TAG-72 PILs were able to adhere to the surface of TAG-72-overexpressing LS174 T human colon cancer cells more effectively than conventional liposomes. Intravenous administration of the anti-TAG-72 PILs, containing plasmids encoding antiangiogenic proteins, significantly inhibited in vivo growth of LS174 T tumors and angiogenesis in the tumor tissues [217]. Anti-STn antibody drug conjugate was generated using CC49 and MMAE, an antimitotic agent that inhibits cell division by blocking the polymerization of tubulin [218]. In the murine model of ovarian cancer OVCAR3, CC49-Br-MMAE-treated mice exhibited an average of a 15.6-day delay in tumor growth and a 40% increase in survival vs. controls. In addition, the anti-STn mAb SF3 MMAE conjugation demonstrated significant tumor growth inhibition in breast and colon STn-expressing tumor xenograft cancer models, without overt toxicity [219].

The most evaluated agent targeting STn was a vaccine (Theratope) that attained phase III clinical trials. This vaccine consists of a synthetic construct of STn disaccharide conjugated to the KLH that has been designed by the biotech company Biomira (Alberta, Canada) [220]. In murine mammary carcinoma models, Theratope immunization induced potent antibody responses that delayed tumor growth [221]. Patients receiving Theratope had a significantly improved survival by 12.1 months and developed anti-STn humoral immune responses [222]. However, no overall benefit of Theratope was observed in a large phase III clinical trial in metastatic breast cancer patients (NCT00003638 [75], and only modest clinical efficacy was achieved in women with metastatic breast cancer who received concurrent endocrine therapy and Theratope (NCT00046371) [76]. Theratope’s lack of efficacy in the phase III clinical trial could be due to the broad variability of STn expression in breast cancer tissues [75,223].

### 3.3. TF Antigen

TF is found in relatively low levels in a lot of normal tissues, but is present in much higher level in carcinomas, and is associated with invasiveness, tumor growth, and high metastatic potential. This is due, at least in part, to increased interaction of cancer cells via TF with members of the endogenous galactoside-binding galectins [224]. TF expression can be determined by different factors including the balance of glycosyltransferases, but also sugar nucleotide transporters and epimerases expression acting synergistically [225]. The TF disaccharide is linked O-glycosidically to the hydroxy amino acid serine or threonine in α-anomeric configuration, but there is an important structural difference between the TF-α disaccharide and the TF-β linked, which can have subtle implications in antibody recognition. Anti-TF antibodies exist in the blood serum of healthy people, originating via cross-reaction with gastrointestinal bacteria, exhibiting TFα-affinity [226]. Although their cancer-related modifications, such as a decrease in anti-TF-specific IgM and increased sialylation, have been linked to cancer and prognosis, suggesting usefulness as biomarkers [227,228], there is still insufficient evidence to validate an anti-TF antibody signature in clinical settings [229].

Classically, TF identification was performed by peanut agglutinin (PNA), although this lectin binds to glycans harboring terminal Galβ found in normal tissues, thereby showing cross-reaction. As a result, great efforts have been made to produce useful specific tools, and several monoclonal antibodies have been developed using different strategies with synthetic or natural TF antigens for mice immunization. Most of them exhibit a cross-reaction between TF-α and TF-β, and controversial results regarding TF expression in cancer have added complexity to the subject. Several antibodies developed in the 1980s and early 1990s have been reviewed in detail by Hanisch and Baldus [230], and we will not add to these reviews. However, we would like to mention some mAbs that show potential therapeutic utility.

The mAb JAA-F11, developed by Rittenhouse-Diakun et al. [231], an IgG3 specific to the TF-α, demonstrated tumor growth inhibition and a decrease in lung metastasis in the breast cancer metastatic model 4T1 [232]. A lack of reactivity with TF-β is important for immunotherapy, considering that this structure is synthetized on glycolipids of normal tissues such as kidney, as well as regenerating respiratory epithelial cells and NK-cells, which can affect tumor growth [233]. Ferguson et al. also demonstrated growth inhibition in a large panel of human breast cancer cell lines [234]. Taken together, the specificity of mAb JAA-F11 and its antitumor growth and antimetastatic effects suggest the potential of this antibody for targeted immunotherapy. As such, humanization of JAA-F11 was performed, obtaining constructs with excellent specificity to TF-α, with lower immunogenicity and able to produce ADCC. This humanized antibody (hJAA-F11) internalized into tumor cells; despite the important demonstration that both naked antibody and amaytansine conjugated antibody (hJAA-F11-DM1) suppressed in vivo tumor progression in a human breast cancer xenograft model in SCID mice [235], no clinical trials have yet been conducted for this antibody.

Using synthetic carbohydrate haptens linked to human serum albumin (HSA) as a carrier protein, Longenecker et al. generated several mAbs [236] with specificities for TF-α, TF-β, and Tn antigens. One of them (170H.82), binding to both forms of TF antigen, reacts with breast, lung, and colon cancers. This antibody labelled with technetium-99m demonstrated improved images in assessing distant metastases in patients with breast cancer [237]. Phase I clinical trials were conducted to study the effectiveness of radiolabeled m170 antibody (Y90 MOAB m170) plus cyclosporine, as well as paclitaxel, in treating patients who have recurrent or refractory metastatic breast cancer (NCT00009763) and metastatic prostate cancer that had not responded to hormone therapy (NCT00009750). Toxicity was limited to marrow suppression and cyclosporine was effective in preventing HAMA reaction [55].

Another strategy targeting TF-antigen showing potent and selective antitumor activity is a peptide specific for TF-α conjugated to the alkylating subunit of the cytotoxin duocarmycin [238]. Duocarmycin is a cytotoxin exhibiting effective antitumor activity in breast and ovarian cancer treatment when conjugated to trastuzumab. The smaller size of the peptides compared to antibodies improved the penetration into solid tumors. An evaluation on different human cell lines confirmed that the peptide-duocarmycin is active in cell lines expressing TF-α.

Regarding vaccine approaches, TF cluster (c)-KLH conjugate vaccine plus QS21 was evaluated in patients with relapsed prostate cancer (NCT00003819) [77]. All doses induced high-titer IgM and IgG antibodies against TF. The results justify the inclusion of TF(c) at a dose of 1 microg as a relevant antigenic target in multivalent phase II vaccine trials in patients in the high-risk minimal disease state.

### 3.4. Parasite Glycans and Cancer Immunotherapy

We have characterized human cancer-associated simple *O*-glycan structures in several parasites [239]. First, we described the presence of Tn and STn antigens in *Echinococcus granulosus*, a cestode parasite causing cystic echinococcosis disease [240], as well as in other species belonging to the two main helminth phyla [241,242,243]. In addition, the sialyl-Tn antigen was detected in *Trypanosoma cruzi*, the protozoan parasite that causes Chagas’ disease [244]. One of the major challenges in the development of an efficient anticancer vaccine is to overcome immune tolerance to tumor-associated antigens (TAA), as well as to the immune evasion strategies developed by tumors [245]. After demonstrating that different TACA are found in parasites, we generated the hypothesis that molecules from these organisms could be useful in developing antitumor vaccines or therapies because TAA from evolutionarily distant organisms should be useful for overriding tolerance problems encountered with human TAA-based cancer therapeutic approaches. We found that immunization with mucin peptides derived from *E. granulosus* could induce antitumor activity by increasing the frequency of activated NK cells and providing splenocytes with the capacity to mediate the killing of cancer cells [246]. In addition, we observed that vaccination with *E. granulosus* antigens from human hydatic cyst fluid (HCF) inhibited colon cancer growth via induction of antitumor immunity [247]. Furthermore, immunization with *T. cruzi* components from epimastigotes (Tce) reduced colon and mammary cancer development in two rat models reproducing human carcinogenesis [248]. Recently, working with a LL/2 lung cancer mouse model, we found that depletion of NK1.1+ cells reduced HCF-induced mouse survival [249]. In addition, oxidative treatment of human HCF with sodium periodate abolished the antitumor activity induced by HCF-vaccination, indicating that glycoconjugates are necessary to induce antitumor responses by *E. granulosus* molecules [249]. Similar results were observed when oxidative treatment with sodium periodate was performed on *T. cruzi* extracts (Freire et al., manuscript in preparation). It is therefore essential to characterize in detail the parasite glycans responsible for antitumor activity induction, and to elucidate the immunobiological mechanisms mediating tumor rejection so as to advance the development of a new type of anticancer treatment.

## 4. Lewis Antigens

Type I and type II Lewis antigens are terminal fucosylated carbohydrate structures belonging to the human histoblood group system, differing only in their glycosidic bonds (Galβ1-3GlcNAc and Galβ1-4GlcNAc, respectively). H1, H2, LewisA (LeA), LewisB (LeB), LewisX (LeX), and LewisY (LeY) are synthetized in exocrine epithelial cells by fucosyltransferases (FUTs) enzymes [250]. These antigens are associated with healthy and pathological conditions and, as cancer-associated antigens, they could be promising targets for novel approaches in personalized medicine [251]. Lewis antigens and their sialylated forms have been largely associated with disease progression and dissemination in cancer patients. CA19.9, which recognize sialyl-LeA (SLeA), has been widely used as a serological tumor marker in gastrointestinal and pancreatic cancers. However, its performance is not accurate enough for clinical use [252]. Although its tissue expression exhibits some conflicting results, probably due to differences in used antibodies, most studies related SLeA detection with worse prognosis [253]. In NSCLC, SLeA and its isomer SLeX are carried by carcinoembryonic antigen, and mediate tumor cells binding to E-selectins on endothelial cells, enhancing metastatic potential [254]. As a therapeutic target, SLeA has attracted interest, and fully human mAbs from individuals immunized with a SLeA–KLH vaccine have been developed, while 5B1 (IgG1) demonstrated improved overall survival in animal models [255]. This antibody, named MVT-5873, was evaluated in pancreatic cancer in combination with gemcitabine and nab-paclitaxel, demonstrating itself to be safe and tolerable [256]; it is being further evaluated in CA19-9 positive malignancies in combination with FOLFIRINOX (NCT02672917). On the other hand, peri-operative MVT-5873 treatment is being evaluated in operable tumors expressing CA19-9, assessing safety and efficacy for improving survival (NCT03801915) [56]. MVT-5873 was also evaluated in combination with MVT-1075 (the same Hu mAb 5B1 conjugated to CHX-A″-DTPA and radiolabelled with 177Lutetium) in a radioimmunotherapy phase II trial in patients with CA19-9-positive tumors. Preliminary results were associated with predicted manageable hematologic toxicities and MVT-1075 demonstrated target accumulation [257]. Another antibody targeting sialyl-di-LewisA is FG129, an IgG1κ and its chimeric form CH129, which was drug-conjugated MMAE or maytansinoid (DM1 and DM4), demonstrating promising results in preclinical models; it is a good candidate for evaluation in clinical trials [258].

Furthermore, mAbs anti-LeY have also been developed and evaluated as potential drugs for cancer therapy. The murine mAb 3S193 (IgG3), generated by immunization with the MCF-7 breast cancer cell line, showed high specificity for LeY antigen [259]. Then, the humanized version hu3S193 was constructed, demonstrating potent immune effector function, with higher ADCC and CDC than its murine counterpart [260]. In clinical studies, it demonstrated itself to be safe, with a strong ability to target tumors, justifying further investigation [57,261] (NCT00084799). However, in a phase II study in gynecological cancers (NCT00617773), the clinical benefit of hu3S193 was modest [58]. Another strategy assessed for this antibody was a second-generation CAR-T anti-LeY evaluated in acute myeloid leukemia, demonstrating feasibility and durable in vivo persistence [262]. Recently, a novel bi-specific antibody was produced, targeting LeY and CD3 (m3s193 BsAb), which showed strong T cell recruiting and antitumor activity in gastric cancer animal models, and suggested potential interest for gastric cancer therapy [263]. Another anti-LeY specific mAb, the chimeric BR96 [264], was drug conjugated to doxorubicin (SGN-15), and it was evaluated in clinical trials in combination with docetaxel for NSCLC (NCT00051571), demonstrating safety and suitability in second- and third-line treatment [59]. In contrast, in metastatic breast cancer, it showed limited clinical antitumor activity and gastrointestinal toxicities [265].

Considering vaccine strategies, SLeA could be an interesting candidate as it is highly detected in several epithelial malignancies, including gastrointestinal, colon and pancreatic cancer, as well as breast cancer and SCLC, but not in counterpart normal tissues. However, considering its structural similarities with other widely expressed blood group-related carbohydrates, cross-reaction may be a major concern. Ragupathi et al. demonstrated that SLeA-KLH conjugate, plus saponin adjuvant, induces high titers of antibodies specific for SLeA, reacting with SLeA-expressing human cancer cells and mediating complement lysis [266]. Evaluated in a pilot study in metastatic breast cancer patients, the results confirmed the safety and immunogenicity of the product, and the authors proposed its inclusion in polyvalent constructs (NCT00470574) [78].

Regarding SLeX, it is a TACA found in different carcinomas (e.g., breast, ovarian, melanoma, colon, liver, lung, and prostate cancer). However, it is also detected on neutrophils, monocytes, and certain T lymphocytes [267]. It is the most important ligand for selectins, particularly E-selectin, which is expressed on the surfaces of endothelial cells, facilitating the leukocyte extravasation to sites of inflammation [268]. Pro-inflammatory cytokines IL-1β, IL-6 and TNF-α could increase the expression of SLeX in cancer cells [269]. Interactions between SLeX of cancer cells and endothelial E-selectin seem to favor the hematogenous metastasis of the overexpressing tumor cells, leading to poor prognosis [270]. Knock down of α-1,3-fucosyltransferase in a colon cancer cell line downregulates SLeX expression and decreases selectin binding and metastatic capacity [271]. Several mAbs specific for SLeX have been developed: FH6 [272], CSLEX-1 [273], AM-3 [274] and KM93 [275], which are useful to identify SLeX in different cancer types. Although several lines of evidence have shown the association between SLeX and the prognosis of patients with breast cancer, colorectal cancer, and NSCLC [276,277], these antibodies have not yet been clinically applied [267]. Strategies with mAbs anti-SLeX have not yet given significant results in cancer treatment, including the recently generated mouse/human chimeric CSLEX-1 antibody [278].

Vaccine development using SLeX has not generated significant contributions either. One strategy to induce immunogenic reaction in melanoma was the injection of SLeX overexpressing polyvalent melanoma cell vaccine (MCV), which consists of three melanoma cell lines (M10-v, M24, and M101) cryopreserved and irradiated [279]. In the melanoma C57BL/6j animal model, better outcomes were observed with high titers of IgM and low titers of IgG, while a high IgG:IgM rate was correlated with worse outcomes [280]. Searching for a switch to a T-dependent response, immunizations with peptides that mimic SLeX proved to be effective in stimulating anti-SLeX specific immune response, and following challenge with Meth-A fibrosarcoma cells, CDC was observed [281]. Besides, small molecule drugs that mimic the structures SLeA and SLeX can potently inhibit their functional binding to selectins and are validated drug targets in the pharmaceutical industry [276]. Among other anticancer strategies, SLeX-modified liposomes have been generated, which are able to inhibit adhesion of cancer cells by competition with ligands and can also be useful for anticancer drug delivering to cancer cells neighboring endothelial cells expressing E-selectin [282,283]. Finally, an interesting observation is that cimetidine, a histamine H2 receptor antagonist inhibiting stomach acid production, dramatically improves survival in patients with colorectal cancer exhibiting high concentrations of SLeX and SLeA in tumor cells [284]. Cimetidine is able to block E-selectin expression on vascular endothelium, and this inhibits adhesion of cancer cells to endothelial cells [285], suggesting that cimetidine can prevent metastasis development by suppression of E-selectin/SLeX and SLeA interaction. Similar results have been observed for gastric cancer [286] and hepatocellular carcinoma [287].

## 5. Polysialic Acid

The glycan modification process, attaching monomeric sialic acid entities onto the nonreducing end of the glycan tree, forms a polysialic acid tail (PSA) synthesized as α2,8-linked homopolymers of Neu5Ac subunits, ranging from 2 to 400 units, on glycoproteins or glycolipids [288]. In humans, a limited number of glycoproteins can undergo polysialylation by enzymes of the α2,8-sialyltransferase (ST8Sia) family, resident in the Golgi apparatus. These enzymes show high substrate specificity for the synthesis of sialylated chains of different lengths (ST8SiaIII for oligo sialic with fewer than seven monosaccharides and ST8SiaII/ST8SiaIV for higher degree of polymerization up to 400 units) [289]. It was demonstrated that the autopolysialylation of ST8SiaII/ST8SiaIV was required for recognition and subsequent polysialylation of substrates such as neuropilin-2 (NRP-2), SynCAM 1, and NCAM [290]. Although PSA is virtually absent in most adult tissues, it is re-expressed during the progression of some malignant tumors, such as neuroblastoma [291], breast cancer [292], laryngeal squamous cell carcinoma [293], pancreatic cancer [294], non-small cell lung cancer [295], and small cell lung cancer [296]. The most important physical feature of PSA is its anti-adhesive effect, favoring invasion and metastasis, but PSA chains also allow for the regulation of signal transduction via influencing the access of ligands to their receptors. For example, PSA-NCAM favors direct binding to fibroblast growth factor 2 (FGF2) and brain-derived neurotrophic factor (BDNF), thus supporting cellular growth and survival [297]. Moreover, polysialylated NCAM, NRP-2, and SynCAM 1 can stimulate various growth factor receptors in cancer cells, promoting oncogenesis via tyrosine kinase pathways, and immune escape of tumor cells by the interaction of PSA with inhibitory siglecs and protecting them from cytotoxicity [298].

The restricted detection of PSA in normal human tissues, in contrast to the abundant expression in some cancers, suggests the rationale for immunological therapy, and several highly specific monoclonal antibodies have been developed: MY.1E12 [299], OL.28 [300], MAB735 [301]. However, no clinical trial has yet been conducted for cancer therapy, although a recent construction and characterization of a chimeric human IgG with MAB735 specificity, drug-conjugated to the tubulin-binding mayntansinoid DM1, demonstrated rapid endocytosis and cytotoxicity against tumor cells expressing PSA, encouraging future evaluations of safety and efficacy in clinical trials [302].

Considering vaccine approaches, immunization with N-propionylated PSA-KLH conjugate plus immunological adjuvant QS21 in patients with small cell lung cancer demonstrated itself to be safe and able to induce high-titer antibody responses, with 10 μg being the lowest optimal immunogenic dose (NCT00004249) [79]. Following these results, N-propionylated PSA was incorporated into a pentavalent vaccine that included four other glycolipids overexpressed in small cell lung cancer (NCT01349647).

## 6. Polyvalent TACA Vaccines

Although monovalent vaccination has resulted in immunologic responses, the heterogeneity of tumor TACA expression suggests that a multivalent approach to generate a broader immune response is preferable. Different strategies have been evaluated for polyvalent TACA vaccine development. A tri-antigenic vaccine, containing Globo H, LeY, and Tn antigens, elicited immune response against each antigen in animal models, and it may result in the recruitment of both humoral and T cell-mediated immune responses against tumors in human patients [303]. A hexavalent vaccine, prepared using GM2, Globo H, LeY, glycosylated MUC-1-32mer and Tn and TF antigens in a clustered formation, KLH conjugated and mixed with QS-21, was evaluated in 30 high-risk prostate cancer patients, inducing significantly high antibody titers in at least two of the six antigens [304]. Another heptavalent vaccine, containing GM2, Globo-H, LeY, Tn-MUC1, and clusters of Tn, STn, and TF, was prepared, mixing TACA individually conjugated to KLH with the adjuvant QS21. Eight of nine patients developed responses against at least three antigens, mainly against Globo-H, Tn-Muc1, Tn, STn, and TF [305]. This vaccine was then improved, producing a unimolecular pentavalent vaccine containing the antigens Globo-H, GM2, STn, TF, and Tn conjugated to KLH, [306], and it was evaluated in a phase I clinical trial in ovarian cancer patients (NCT01248273) [80]. This unimolecular pentavalent vaccine demonstrated itself to be safe and immunogenic, and all but 1 of the 24 vaccinated patients (96%) exhibited antibody responses against at least one of these antigens, and 20 patients (83%) responded against at least three antigens after vaccination. The longest progression-free survival was observed in a patient who developed an IgM response to all five antigens present in the vaccine, while the only patient whose immune response was null to any of the antigens had the shortest progression-free survival. On the other hand, this strategy of polyvalent vaccines has been evaluated in other tumors such as breast cancer with Globo-H-GM2-Lewis-Y-MUC1-32(aa)-STn(c)-TF(c)-Tn(c)-KLH conjugate + QS21 (NCT00030823); and SCLC with GD2L, GD3L, Globo H, fucosyl GM1, and N-propionylated polysialic acid -KLH conjugate + OPT-821 (NCT01349647). A pilot study of a polyvalent vaccine-KLH conjugate + OPT-821, given in combination with bevacizumab, was conducted in patients with recurrent epithelial ovarian, fallopian tubes, or primary peritoneal cancer who are in second or greater clinical remission (NCT01223235), demonstrating it to be safe.

## 7. CAR-T Cells

CAR-T cells combine the specificity of an antibody with the cytolytic capacity of T cells in an MHC-independent manner [307]. These therapies have proved successful for hematological malignancies, but in solid tumors, multiple challenges need to be overcome, including identification of new tumor-associated antigens, the limited infiltration of CAR-T cells to tumor sites, and the immunosuppressive effect of the tumor microenvironment [308]. In this way, some TACA have demonstrated usefulness for antitumor CAR-T engineering, and the most notable developments have been achieved with anti-GD2, anti-MUC1/MUC1-Tn, and anti-STn.

### 7.1. CAR-T Cells Targeting GD2

At the beginning of the century, Rossig et al. published one of the first CAR-targeting GD2, demonstrating in vitro that first-generation CAR-T cells derived from 14.G2a antibodies recognized and lysed GD2 positive cancer cells in an antigen-specific manner [309]. The first clinical trial used Epstein Barr Virus (EBV)-specific T cells, engineered with first generation CAR targeting GD2, recruited 11 patients with neuroblastoma. This therapy was demonstrated to be safe, and four out of eight patients had evidence of tumor necrosis or regression, including a sustained completed remission [310]. A subsequent study of the same group reported the long-term analysis of the mentioned trial in a total of 19 patients, including the first cohort, demonstrating persistence of circulating CAR-T cells for at least four years [311]. A third generation of CAR-T cells targeting GD2, incorporating CD28 and OX40 signaling domains (GD2-CAR3 T cells), administered after lymphodepletion by cyclophosphamide and fludarabine, and in combination of the PD-1 inhibitor pembrolizumab, demonstrated itself to be safe, but with modest clinical responses [312]. A humanized scFv derived from the anti-GD2 K666 antibody was used for a second-generation CAR-T production, which was evaluated in a phase I study in refractory/relapsed neuroblastoma patients (NCT02761915), but no patients presented objective clinical response, although three of the six patients who received higher amounts of CAR-T cells in lymphodepleting conditions showed antitumor activity and signs of immune activation without on-target off-tumor neurotoxicity [313]. Recently, Yu et al. published preliminary results of a phase I clinical study (NCT02765243) using autologous fourth generation CAR-T cells engineered to expresses an anti-GD2 derived from humanized mouse 3F8 scFv, a CD3-zeta domain, a 41BB domain, a CD28 extracellular and intracellular domain and an inducible caspase 9 domain, named 4SCAR-GD2 T cells [314]. The inducible caspase 9 domain allows the elimination of CAR-T cells by administration of the agent AP1903 if necessary. For the first 12 enrolled patients, this therapy demonstrated an antitumor effect and manageable toxicities, indicating its potential to benefit children with refractory and/or recurrent neuroblastoma. Interestingly, it was recently observed that myeloid-derived suppressor cells (MDSC) increased in peripheral blood of neuroblastoma patients after administration of CAR-T cells targeting GD2 in the case of relapse and loss of response. Moreover, in patients treated with CAR-T cells, the frequency of circulating polymorphonuclear (PMN) MDSCs inversely correlates with the levels of CAR-T cells, resulting in more elevated levels in patients who did not respond or lost response to CAR-T treatment. This study highlights the prognostic value of PMN-MDSC and places it as an interesting target for therapeutic agents in order to protect CAR-T cells from detrimental actions (NCT03373097) [315].

Gene engineering of natural killer (NK) cells rather than T cells could be a safer alternative to targeting GD2-positive cancers because they have a shorter life span. CAR-NKT cells co-expressing a GD2-specific CAR and Il-15 have shown encouraging results in preclinical trials using several xenograft models of neuroblastoma. This CAR-NKT demonstrated superior in vivo persistence, tumor infiltration, and antitumor activity in a metastatic neuroblastoma mouse model, while no evident toxicity was observed, providing reasons for a first in-human CAR-NKT cell clinical trial (NCT03294954) [316]. Preliminary results from three patients demonstrate that this treatment is safe and suggest that CAR-NKT cells can be expanded to a clinical scale [317].

In addition to neuroblastoma, CAR-T cells targeting GD2 have been evaluated for treatment of other cancers expressing this ganglioside. Third-generation CAR-T cells derived from the anti-GD2 scFv 14.G2a showed antitumor activity in murine xenograft cancer models of melanoma [318], and Yu et al. demonstrated that second generation CAR-T cells with the same anti-GD2 specificity induced rapid tumor regression in melanoma patient-derived xenograft models [319]. However, although promising, this therapy exhibits some limitations in melanoma patients, including off-target toxicity and resistance, reviewed by Soltantoyeh et al. [320]. This is an active development field and there are currently 16 ongoing phase I/II clinical trials using CAR-T targeting GD2 in different cancers to improve clinical outcomes (NCT03373097, NCT04099797, NCT04539366, NCT04430595, NCT01953900, NCT03635632, NCT03423992, NCT04196413, NCT04429438, NCT03721068, NCT01822652, NCT04637503, NCT03294954, NCT03356782, NCT00085930, NCT04433221). Table 4 shows selected clinical trials of anti-TACA CAR-T cells.

### 7.2. CAR-T Cells Targeting MUC1/MUC1-Tn

CAR-T cell strategies targeting MUC1/MUC1-Tn have demonstrated encouraging results in preclinical studies using different in vivo cancer models. Since Wikie et al. demonstrated, for the first time, that anti-MUC1 CAR-T cells generated from the scFv of HMFG2 antibody present in vivo antitumor activity in a breast cancer xenograft [323], a growing number of reports show that this is an active and promising clinical investigation field. A CAR-T MUC1-Tn constructed by the scFv of 5E5 mAb demonstrated potent antitumor activity in preclinical xenograft models of T cell leukemia and pancreatic cancer [324]. Moreover, an interesting study demonstrated that second-generation CAR-T cells targeting MUC1 (derived from HMFG2 antibody) could efficiently suppress tumor growth in a patient derived xenograft (PDX) mouse model of NSCLC expressing MUC1 [325]. A phase I clinical trial using two different constructs of anti-MUC1 CAR-T cells derived from the SM3 antibody exhibited serum cytokine responses and no side-effects in a patient with metastatic seminal vesicle cancer [326]. Indeed, one of the CAR-T MUC1s used in this study effectively caused tumor necrosis, confirming the effectiveness of this therapeutic approach. Furthermore, CAR-T with the same specificity (SM3 scFv) but engineered PD-1 KO was evaluated in NSCLC (NCT03525782), demonstrating that the treatment is safe and well tolerated [321].

Another study explored the antitumor activity of two CAR-T MUC1 cells of second and fourth generation, including the last one the expression of IL-22. Both CAR-T cells, derived from an anti-MUC1 scFv (VH: HMFG2, VL: SM3), exhibited specific antitumor activity in a head and neck squamous cell carcinoma xenograft mouse model [327]. Considering that the immunosuppressive tumor microenvironment is one of the challenges that must be overcome in CAR-T therapy, a strategy for remodeling it could be CAR-T cells’ anti-MUC1 derived from HMFG2 antibody and expressing the costimulatory receptor TR2.41BB [328]. In vivo experiments show that this approach enhances the expansion, persistence, and antitumor activity of the anti-MUC1 CAR-T cells in a triple-negative breast cancer xenograft mouse model. Recently, a second-generation CAR γδ T cell using anti-MUC1-Tn scFv PG926 was reported, displaying significantly enhanced antigen-specific antitumor potency, both in vitro and in vivo, in a gastric carcinoma human model [329]. Another CAR-T MUC1 currently in clinical evaluation is huMNC2-CAR44, targeting MUC1*, the extra cellular domain of the cleaved form of MUC1, present on a large percentage of solid tumors, including breast cancer (NCT04020575). The investigators propose to evaluate safety and preliminary antitumor activity in patients with metastatic MUC1*-positive breast cancer [322]. Moreover, a multicentric clinical trial is evaluating a CART-TnMUC1 immunotherapy in solid tumors (triple-negative breast cancer, epithelial ovarian cancer, pancreatic cancer, and non-small cell lung cancer) and TnMUC1 positive multiple myeloma, designed to identify the dose and regimen at which CART-TnMUC1 cells can be safely administered intravenously following lymphodepletion (NCT04025216) [330]. In a second expansion phase enrolling 72 more patients, efficacy is due to be assessed. Finally, third-generation CAR-T cells targeting MUC1 but also multiple other targets (PSCA, TGFβ, HER2, Mesothelin, LeY, GPC3, AXL, EGFR, Claudin18.2, or B7-H3) were constructed, and clinical studies will be performed to test the anticancer function of these individual or combination CAR-T cells for immunotherapy in human cancer. This is an interesting strategy that could help overcome cancer heterogeneity, and it is currently being evaluated in lung cancer in an ongoing clinical trial (NCT03198052).

### 7.3. CAR-T Cells Targeting STn

The first CAR against STn was reported by Hombach et al. more than 20 years ago, and few years later, McGuiness et al. demonstrated that first generation of CAR-T cells targeting STn presented antitumoral activity in two murine xenograft models of endometrial and colon adenocarcinoma [331,332]. More recently, a dual-specific CAR, containing an antagonist anti-CD30 scFv derived from the HRS3 antibody and an anti-STn scFv, presented high levels of cytotoxic activity on LS-C colon cancer cells in vitro [333]. Antitumor activity of second-generation STn CAR-T cells, constructed with the humanized antibody CC49, was evaluated in xenograft models of peritoneal ovarian cancers [334]. In this study, regional intraperitoneal delivery of TAG72-BBζ CAR-T cells significantly reduced tumor growth and improved overall survival of mice. The strategy of targeting multiple antigens to improve effectiveness was evaluated by Shu et al. by generating dual CAR-T cells targeting STn and CD47 (overexpressed in multiple cancer types and expressed on many normal cells). In order to minimize the killing of heathy cells, they designed a truncated anti-CD47 CAR, devoid of intracellular signaling domain (ΔCD47), and a second generation anti-STn CAR derived from the antibody CC49 [335]. The CD47 CAR facilitates binding to CD47 positive cancer cells, increasing the chances of TAG72 positive cancer cells’ elimination via STn CAR. They used two ovarian cancer xenograft models, OVCAR-3 and MESOV, with high and low expressions of STn, respectively, but both cell lines expressed high levels of CD47. In these preclinical models, single STn CAR-T cell therapy was effective in delaying tumor growth in the OVCAR-3 cancer model, but it was unable to delay tumor growth in the MESOV cancer model. However, tumor growth was delayed in the MESOV cancer model using dual STn + ΔCD47 CAR-T cells [335].

In 2017, Hege et al. published the results of the first human clinical trials of CAR-T cells in the treatment of solid tumors, reporting data from two studies using first-generation CAR-T cells targeting STn in metastatic colorectal cancer patients [336]. Both studies lack clear evidence of on-target/off-tumor toxicity, but no clinical responses were observed. The authors also reported the trafficking to tumor tissue of CAR-T cells in one of three patients, and some patients induced an interfering antibody to the STn binding domain of humanized CC49, associated with rapid clearance of subsequent CAR-T cell infusions, suggesting the potential benefit of incorporating co-stimulatory domains in the CAR design.

## 8. Concluding Remarks and Perspectives

The glycan pattern expressed in cancer cells has a significant effect on tumor behavior. Different TACAs are established biomarkers, with recognized usefulness in clinical oncology. TACA overexpression on a wide range of cancer cells is an attractive target for immunotherapy developments. The main conclusions emerging from this review highlight that immunotherapy strategies targeting TACAs can be approved for cancer therapy, such as anti-GD2 antibodies in neuroblastoma treatment. While some TACAs are excellent tumor biomarkers, the development of immunotherapies targeting them is a major difficulty. Although some progress has been made in this field, no TACA-based vaccine has been approved thus far by the FDA. Thus, there remains a great need to improve the immunogenicity of TACAs, which is critical for cancer therapy, because they are unable to induce T cell-dependent immune responses by themselves. As well as improving strategies for TACA coupling to peptide epitopes that are able to be recognized by T cells, as well as novel polyvalent vaccine development and the characterization of strong and safe immune adjuvants, a promising approach could be vaccine developments based on chemically modified TACAs [337]. Some examples include an oxime-linked Tn analogue [338], a fluoro-substituted Sialyl-Tn [339], and a MUC1-β-TF [340].

Monoclonal antibodies represent one the most effective classes of biopharmaceuticals for cancer treatment. A variety of diverse and innovative approaches continue to be developed, based on the generation/modification of anti-TACA antibodies to obtain new and more effective antibody-based therapies (Figure 1). In addition to the usefulness of different naked mAbs in immunotherapy, other emerging promising approaches are bi-specific antibodies, antibody–drug conjugates, and CAR-T cells. Antibody engineering is a great opportunity for the discovery and development of novel therapeutic tools. One of the main limitations of anti-glycan antibodies is their low affinity for antigen recognition, but genetic engineering allows us to increase this affinity, improving antitumoral activity. For example, by using a yeast surface display, novel antibodies with increased affinity and better specificity were generated from mAb anti-SLeA, improving binding to human pancreatic and colon cancer cell lines and complement-dependent therapeutic efficacy [341]. This strategy could allow for improvements to the antitumor therapeutic potential of different anti-TACAs antibodies. In addition to antigen specificity, mAbs comprise Fc-mediated effector functions, such as the ADCC- and ADCP-activating Fcγ receptor (FcγR), which are of major importance for the therapeutic efficacy of mAbs. FcγR-mediated effector functions, optimized by strategies such as point mutations, altered glycosylation patterns, combinations of different Fc subclasses (cross isotypes), and Fc-truncation of the mAb, can address important improvements regarding clinical safety and efficacy [342]. Recently, the Fc of an anti-sialyl-di-LeA antibody was modified to incorporate mechanisms of tumor death in murine anti-glycan antibodies. Murine IgG3 mAbs targeting glycans often induce direct cell killing in the absence of immune effector cells, or complement, via a proinflammatory mechanism resembling oncotic necrosis. By identifying the key residues within mouse IgG3 responsible for noncovalent Fc interactions and transferring them into human IgG1, a chimeric human IgG1 was created, which exhibited increased in vitro and in vivo antitumor activity [343]. Following this strategy, the antitumor activity of different anti-TACA mAbs can also be improved.

New perspectives are ready for exploration in cancer therapy; for example, by removing some glycans from the cancer cell surface, we can improve its susceptibility to immunotherapy. Considering sialic acid residue involvement in tumor immune-evasion mechanisms, the laboratory of Carolyn Bertozzi developed a strategy to eliminate sialic acids of HER2+ breast cancer cells. Using mAb trastuzumab (anti-HER2), they developed an antibody–sialidase conjugate that was able to enhance cancer cell susceptibility to ADCC by selective desialylation of the glycocalyx of the targeted tumor cell [344]. Sialidase conjugation to trastuzumab also enhanced ADCC in tumor cells expressing moderate levels of HER2, suggesting a therapeutic strategy for cancer patients with lower levels of HER2 or trastuzumab resistance [344]. More recently, the authors found that removal of sialoglycans from cancer cells by this strategy could improve the antitumor immune response in mice through a siglec-E-dependent mechanism [345].

Many approaches that are effective in cancer treatment are based on the aberrant glycosylation of tumor cells. The expanding knowledge of cancer glycobiology and access to novel biotechnological tools determine that dynamic achievements in the field have growing clinical impact. In this review, we only analyzed aspects related to TACA targeting. Whatever the treatment options, they must be used in synergy with other anti-tumor strategies for the adequate personalization of therapy.

## Figures and Tables

**Figure 1 cancers-14-00645-f001:**
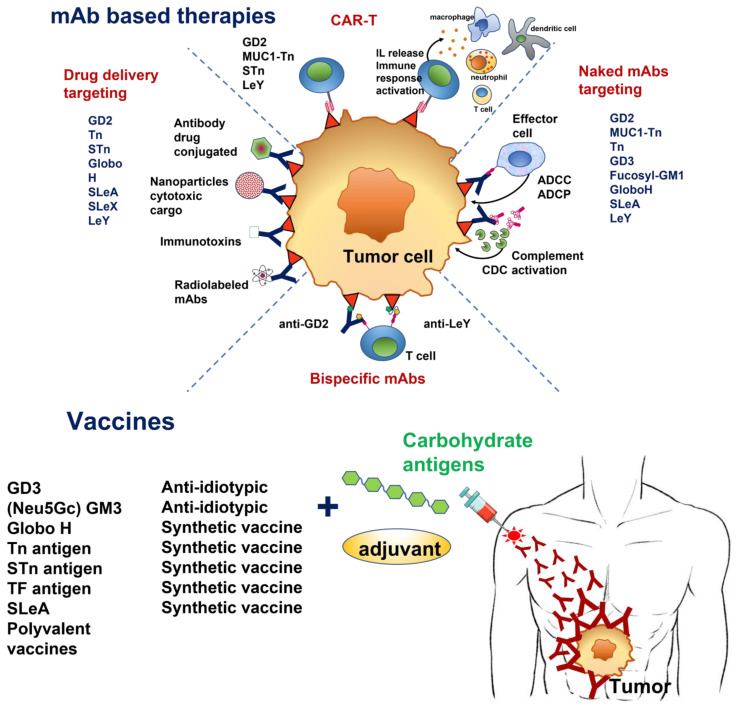
Schematic representation of antitumor strategies based in antibodies and vaccines targeting TACAs (Tumor Associated Carbohydrate Antigens). mAbs (monoclonal antibodies); CAR-T (Chimeric antigen receptor-T); STn (sialyl Tn); SLeA (sialyl Lewis A); SLeX (sialyl Lewis X); LeY (Lewis Y); ADCC (Antibody-Dependent Cell-mediated Cytotoxicity); ADCP (Antibody-Dependent Cellular Phagocytosis); CDC (complement-dependent cytotoxicity).

**Table 1 cancers-14-00645-t001:** Main tumor glycans targeted for cancer therapy.

Tumor Glycan	Glycan Type	Structure *	Type of Cancer	Function in Cancer	Therapeutic Strategy
GD2	Ganglioside	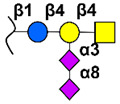	GliomaMelanomaNeuroblastomaRetinoblastoma	Cell proliferationMotilityApoptosis resistance	mAbs DinutuximabNaxitamabBispecific-AbsCAR-T cells
GD3	Ganglioside	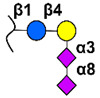	GliomaMelanomaBreast, Lung	Cell growthInvasion	mAb huR24Anti-idiotypic mAb BEC2
GM3(Neu5Gc)	Ganglioside	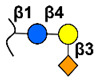	MelanomaLungColonBreast	Cell growthMetastasis	mAb 14F7hTAnti-idiotypic mAb RacotumomabSynthetic vaccine
Fucosyl-GM1	Ganglioside	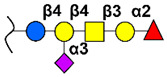	SCLC	The role in cancer is unclear	Human mAbBMS-986012
Globo H	Globoside	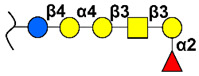	Colon, Ovary, Breast, Prostate, Gastric, Lung, Endometrial, Pancreatic	AngiogenesisImmunosuppression	VaccinemAb OBI-888ADC OBI-999
Tn antigen	O-GalNAc mucin-type	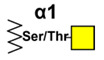	Carcinomas	MetastasisImmuno-suppression	AntibodiesVaccineCAR-T cells
STn antigen	O-GalNAc mucin-type	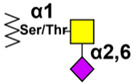	Carcinomas	MetastasisImmuno-suppression	AntibodiesVaccineCAR-T cells
TF antigen	O-GalNAc mucin-type	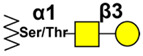	Carcinomas	Cell growthAdhesion	Vaccine
SLeA	GlycoproteinGangliosides	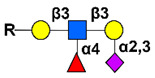	Breast, Colon,Gastric, Lung,Ovarian, Pancreas	Metastasis	Abs MVT-5873177Lu- MVT-1075Vaccine
SLeX	GlycoproteinGangliosides	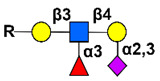	Colon, breastmelanoma, lung, liverovary, pancreas	InvasionMetastasis	SLeX -liposomesanticancer drug delivery
LeY	GlycoproteinGangliosides	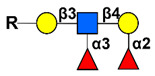	Lung, Ovarian,Fallopian tube,Breast	Metastasis	mAb hu3S193m3s193 BsAbDCA SGN-15CAR-T cells
Poly-sialic acid	GlycoproteinGlycolipids	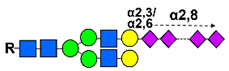	NeuroblastomaBreast, PancreasLung	MetastasisCell growth	Abs MY.1E12,OL.28, MAB735Vaccine

* Colored symbols represent the following monosacharides: yellow square: N-Acetylgalactosamine (GalNAc); blue square: N-Acetylglucosamine (GlcNAc); violet diamond: Sialic acid (NeuAc); orange diamond: N-glycolylneuraminic acid (Neu5Gc); yellow circle: Galactose (Gal); blue circle: Glucose (Glc); green circle: Mannose (Man); red triangle: Fucose (Fuc).

**Table 2 cancers-14-00645-t002:** Selected clinical trials of antibodies targeting tumor glycans.

Target	Drug Candidate	Phase	Cancer Type	Status Time Period	CT ID Refs
GD2					
	hu 14.18 (Dinutuximab)	III	High risk Neuroblastoma	Active, not recruiting October 2010–September 2021	NCT00026312 [36]
	hu3F8 (Naxitamab) + GM-CSF	II	High risk Neuroblastoma	Recruiting 4/2018–11/2027	NCT03363373
	hu3F8 + GM-CSF	I/II	High risk Neuroblastoma	Active, not recruiting 12/2012–12/2023	NCT01757626 [43,44]
	Naxitamab hu3F8 + GM-CSF + irinotecan and temozolomide	II	Recurrent Neuroblastoma	Recruiting 11/2021–10/2026	NCT04560166
	Dinutuximab hu 14.18 + GM-CSF	II	Lung cancer Osteosarcoma	Active, not recruiting 11/2015–10/2021	NCT02484443
	hu3F8 + GM-CSF	II	Osteosarcoma	Recruiting 7/2015–7/2022	NCT02502786 [45]
	hu14.18-IL2	II	Melanoma	Completed 12/2007–11/2019	NCT00590824 [46,47]
	I 131 mAb 3F8 + bevacizumab	I	Neuroblastoma	Completed 8/2006–9/2015	NCT00450827
	I 131 mAb 3F8	I	Central nervous system, Leptomeningeal metastases	Active, not recruiting 1/2006–1/2022	NCT00445965
	Hu3F8-Bispecific antibody (GD2-CD3)	I/II	Neuroblastoma Osteosarcoma Solid Tumors	Recruiting 2/2019–2/2022	NCT03860207 [48]
	Anti-GD2/CD3 bispecific antibody (Nivatrotamab)	I/II	SCLC	Recruiting 8/2021–12/2024	NCT04750239
GD3					
	PF-06688992 Anti-GD3/drug conjugated	I	Stage III/IV Melanoma	Completed 5/2017–1/2020	NCT03159117
	Human Chimeric Ab KW2871 (ecromeximab)	I/II	Stage IV Melanoma	Terminated 11/2004–2/2015	NCT00199342 [49]
	Human Chimeric Ab KW2871 (ecromeximab) + high dose IFN-α2b	II	Metastatic Melanoma	Completed 3/2008–2/2018	NCT00679289 [50]
fucosyl-GM1					
	BMS-986012 + platinum + etoposide	I/II	SCLC	Active, not recruiting 11/2016–4/2021	NCT02815592
	BMS-986012	I/II	Relapsed/refractory SCLC	Active, not recruiting 11/2014–6/2022	NCT02247349 [51]
	BMS-986012 + carboplatin + etoposide + nivolumab	II	Extensive-stage SCLC	Recruiting 3/2021–9/2024	NCT04702880
Globo H					
	OBI-888	I/II	Advanced and metastatic solid tumors	Recruiting 5/2018–12/2022	NCT03573544
	OBI-999 immunotoxin	I/II	Locally advanced solid tumors	Active, not recruiting 12/2019–12/2023	NCT04084366
Tn antigen					
	Anti- TA-MUC1 (PankoMab-GEX™)	I	Solid tumors	Completed 11/2009–5/2021	NCT01222624 [52]
	Anti- TA-MUC1 (PankoMab-GEX™)	II	Recurrent Ovarian cancer, Fallopian cancer, Peritoneal cancer	Completed 9/2013–10/2020	NCT01899599 [53]
	Anti- TA-MUC1 (Gatipotuzumab) + anti-EGFR (Tomuzotuximab)	I	Solid tumors	Completed 11/2017–7/2021	NCT03360734 [54]
TF antigen					
	yttrium Y 90-m170 cyclosporine paclitaxel	I	Breast cancer	Unknown 3/2001–9/2013	NCT00009763 [55]
	yttrium Y 90-m170 cyclosporine paclitaxel	I	Prostate cancer	Unknown 3/2001–9/2013	NCT00009750 [55]
LeA					
	hu mAb-5B1 (MVT-5873) + FOLFIRINOX	1	Pancreatic cancer or CA19-9 positive malignancies	Recruiting 1/2016–1/2023	NCT02672917
	hu mAb-5B1 (MVT-5873)	2	Operable tumors expressing CA19-9	Recruiting 11/2019–12/2024	NCT03801915 [56]
LeY					
	hu3S193	1	SCLC	Completed 2/2004–6/2015	NCT00084799 [57]
	hu3S193	2	Fallopian tube, Ovarian cancer, Primary peritoneal cancer	Completed 5/2008–6/2012	NCT00617773 [58]
	SGN-15 + docetaxel	2	NSCLC	Completed 1/2003–10/2011	NCT00051571 [59]

**Table 3 cancers-14-00645-t003:** Selected clinical trials of anti-cancer vaccines based on tumor glycans.

Target	Drug Candidate	Phase	Cancer Type	Status/ Time Period	CT ID Refs
GD3					
	BEC2 + BCG	III	Small cell lung cancer	Completed September 1998–April 2010	NCT00037713 [69]
	BEC2 + BCG	III	Lung cancer	Completed September 1999–July 2012	NCT00006352 [70]
	BEC2 + BCG	III	Lung cancer	Completed 3/1998–3/2012	NCT00003279 [71]
Glycolyl GM3					
	Racotumomab	I	Pediatric tumors	Completed 2/2011–7/2015	NCT01598454 [72]
	Racotumomab	II	High risk Neuroblastoma	Active, not recruiting 11/2016–9/2022	NCT02998983
	Racotumomab	II	NSCLC	Completed 9/2009–7/2014	NCT01240447
	Racotumomab	III	NSCLC	Unknown 9/2010–7/2016	NCT01460472
Globo H					
	OPT-822/OPT-821 + cyclophosphamide	II	Metastatic breast cancer	Completed 12/2011–9/2020	NCT01516307 [73]
	OBI 822(adagloxad simolenin)/OBI-821	III	Triple negative breast cancer	Recruiting 12/2018–12/2027	NCT03562637
Tn antigen					
	MAG-Tn3 + AS15	I	Breast cancer	Active, not recruiting 2/2015–12/2021	NCT02364492 [74]
STn antigen					
	STn/KLH (THERATOPE^®^)	III	Metastatic breast cancer	Completed 1/1999–3/2013	NCT00003638 [75]
	STn/KLH (THERATOPE^®^)	II	Metastatic breast cancer	Completed 8/2002–1/2008	NCT00046371 [76]
TF antigen					
	TF(c)-KLH + QS21	1	Prostate cancer	Completed 6/1998–3/2013	NCT00003819 [77]
Lewis					
	sialyl Lewis A-KLH + QS21		Metastatic breast cancer	Completed 3/2007–1/2020	NCT00470574 [78]
Polysialic acid					
	Polysialic acid-KLH + QS21	II	Small cell lung cancer	Completed 8/1998–6/2013	NCT00004249 [79]
Polyvalent vaccines					
	Bivalent vaccine (GD2L/GD3L) + OPT-821	I/II	Neuroblastoma	Active, not recruiting 5/2009–5/2023	NCT00911560 [67,68]
	Trivalent vaccine GM2/GD2L/GD3L-KLH conjugated + OPT-821	II	Sarcoma	Completed 6/2010–3/2017	NCT01141491 [68]
	Globo-H-GM2-STn-TF-Tn-KLH- QS21	I	Ovarian cancer	Completed 11/2010–03/2017	NCT01248273 [80]
	Globo-H-GM2-LeY-MUC1-32(aa)-STn(c)-TF(c)-Tn(c)-KLH conjugate vaccine- QS21		High risk breast cancer	Completed 03/2001–12/2015	NCT00030823
	GD2L, GD3L, Globo H, fucosyl GM1, and N-propionylated polysialic acid -KLH + OPT-821	I	Small cell lung cancer	Completed 05/2011–01/2016	NCT01349647

**Table 4 cancers-14-00645-t004:** Selected clinical trials of CAR-T cells targeting tumor glycans.

Target	Drug Candidate	Phase	Cancer Type	Status/Period	CT ID Refs
GD2					
	1RG-CART + Cyclophosphamide + Fludarabine	I	Relapsed or refractory Neuroblastoma	Completed February 2016–December 2020	NCT02761915 [313]
	4SCAR-GD2	I	Neuroblastoma	Suspended January 2016–December 2022	NCT02765243 [314]
	GD2-CART01	I/II	Neuroblastoma and GD2 positive solid tumors	Recruiting May 2018–December 2027	NCT03373097 [315]
	GD2-CAR NKT cells (GINAKIT)	I	Neuroblastoma	Recruiting 1/2018–8/2034	NCT03294954 [316,317]
MUC1/MUC1-Tn					
	MUC1-CAR+/PD-1- T cells	I/II	NSCLC	Recruiting 2/2018–1/2022	NCT03525782 [321]
	huMNC2-CAR44	I	Metastatic breast cancer	Active 1/2020–1/2035	NCT04020575 [322]
	CART-TnMUC1	I	NSCLC, Ovarian cancer, Fallopian tube cancer, Triple negative breast cancer, Multiple myeloma, Pancreatic ductal adenocarcinoma	Recruiting 10/2019–10/2036	NCT04025216
	CAR-T cells targeting PSCA, MUC1, TGFβ, HER2, Mesothelin, LeY, GPC3, AXL, EGFR, Claudin18.2, or B7-H3	I	Lung cancer	Recruiting 7/2017–8/2023	NCT03198052

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
