# Peer review of "Targeting Tumor Glycans for Cancer Therapy: Successes, Limitations, and Perspectives"

_cancers, 2022, doi:10.3390/cancers14030645_

Round 1
Reviewer 1 Report
This is a very comprehensive and thoroughly researched piece of work, very detailed. It is clearly written and presented. It presents a detailed synthesis of the state of the art regarding the past and ongoing clinical trials targeting cancer associated glycans, and their potential. I have only very minor suggestions for improvement:
Throughout, it would be helpful to refer to 'cancers' rather than 'tumours', if you mean 'cancers'. Obviously many tumours are benign. This paper appears to be focused on malignant tumours -- cancers -- only.
The legend to Table 1 should define the monosaccharide abbreviations, eg Gal, GalNAc etc.
In Table 1, the linkages (eg beta 3 etc) are given for some structures (eg SLeA and LeY) but not for others (eg Tn and sTn). They should be given for either all, or none -- ie be consistent. Seeing the linkages is helpful, so I would recommend including them for all. Moreover, for the naive reader, the symbols for the linkages should be explained in the legend.
In Tables 1 and 2, and in the text, there are some inconsistencies in how some glycans are described. For example, in Table 1 you write 'Tn' and 'TF', in Table 2 'Tn antigen' and 'TF antigen'. In Table 1 'Sialyl Tn' but 'SLeA'. Choose a standard form and be consistent throughout.
L101 'based on variations in the glycan buildup' -- I don't understand what that means. Can you clarify.
L154 explain what 'bispecific antibodies' are
L198-9 'it has been demonstrated that overexpression of GD3 and GD3S plays a key role in tumorigenicity' -- what do you mean by 'tumorigenicity' here? That they are associated with cancers but not normal? That they are associated with poor prognosis? Explain
L394 (truncated O glycans) 'are significantly associated with tumour progression through various mechanisms' -- can you explain what mechanisms? As far as I am aware, these glycans have been frequently reported to be associated with cancers, and sometimes with poor prognosis, but the mechanisms by which they might contribute to cancer progression are largely unknown or poorly understood. If you are aware of evidence for their function in cancer progression, it would be helpful to the reader to present it.
L427 'built on different backbones such as S*S*S*, S*T*T* or T*T*T* (*=GalNAc). You need to explain what S and T are -- I assume serine and threonine?
L530 'STn is not expressed by normal cells and tissues' and L596 'TF, like Tn antigen, is found almost exclusively in tumours'. Check whether this is correct. I think Tn antigen is rarely found in normal tissues, commonly in some cancers. TF and STn, I think, are found in relatively low levels in alot of normal tissues, but are present in much higher levels in many cancers. But I don't think it is accurate to say that they are not present in normal. Also avoid using the term 'expressed' when referring to glycans, because, of course, they are not gene products so are not 'expressed', they are synthesised.
L534 'leading to more aggressive cancer behaviour' -- define what is meant by 'aggressive cancer behaviour'. Do you mean, for example, reduced survival?
L636 'human serum albumin' is abbreviated as 'HAS' -- should it be HSA?
L693 explain what 'FUT enzymes' are
Section 5, there are several instances where 'polysialilation' and 'polysialilated' are mis-spelled - they should read polysialylation and polysialylated.
Author Response
Response to Reviewer 1 Comments
Point 1: Throughout, it would be helpful to refer to 'cancers' rather than 'tumours', if you mean 'cancers'. Obviously many tumours are benign. This paper appears to be focused on malignant tumours -- cancers -- only.
Response 1: We agree with the reviewer, and we replace ´tumor´ by ´cancer´ whenever it was appropriate.
Point 2: The legend to Table 1 should define the monosaccharide abbreviations, eg Gal, GalNAc etc.
Response 2: We thank the reviewer for this comment, and we defined monosaccharide abbreviations at the Table 1 legend, which certainly improve reader comprehension.
Point 3: In Table 1, the linkages (eg beta 3 etc) are given for some structures (eg SLeA and LeY) but not for others (eg Tn and sTn). They should be given for either all, or none -- ie be consistent. Seeing the linkages is helpful, so I would recommend including them for all. Moreover, for the naive reader, the symbols for the linkages should be explained in the legend.
Response 3: We agree with the reviewer regarding linkages, and they were included for all structures.
Point 4: In Tables 1 and 2, and in the text, there are some inconsistencies in how some glycans are described. For example, in Table 1 you write 'Tn' and 'TF', in Table 2 'Tn antigen' and 'TF antigen'. In Table 1 'Sialyl Tn' but 'SLeA'. Choose a standard form and be consistent throughout.
Response 4: Thank you for pointing this out. We carefully revised all the manuscript and all inconsistencies have been corrected.
Point 5: L101 'based on variations in the glycan buildup' -- I don't understand what that means. Can you clarify.
Response 5: We agree with the reviewer that the sentence is somewhat confusing. We clarify as follow: “Over 300 different GSLs are synthesized by enzymes localized in the ER and Golgi apparatus, integrating signaling components, assembly of glycosylating machinery and GSL trafficking, although not completely understood to date [23].”
Point 6: L154 explain what 'bispecific antibodies' are
Response 6: We explain what bispecific antibodies are as follow: “… vaccines, chimeric antigen receptor T-cells, as well as bispecific antibodies (BsAbs, antibodies with two binding sites directed at two different antigens). “
Point 7: L198-9 'it has been demonstrated that overexpression of GD3 and GD3S plays a key role in tumorigenicity' -- what do you mean by 'tumorigenicity' here? That they are associated with cancers but not normal? That they are associated with poor prognosis? Explain
Response 7: We agree with the reviewer that the term ´tumorigenicity´ can be misunderstood. We explain better as follow: “. It has been demonstrated that overexpression of GD3 and GD3S in glioblastoma stem cells plays a key role in tumorigenesis by expression of stemness genes and self-renewal potential [57] and the enzyme is frequently overexpressed in other tumors such as breast cancer, melanoma, lung cancer, and hepatocarcinoma, leading to be proposed as novel drug targets in cancer [58].”
Point 8: L394 (truncated O glycans) 'are significantly associated with tumour progression through various mechanisms' -- can you explain what mechanisms? As far as I am aware, these glycans have been frequently reported to be associated with cancers, and sometimes with poor prognosis, but the mechanisms by which they might contribute to cancer progression are largely unknown or poorly understood. If you are aware of evidence for their function in cancer progression, it would be helpful to the reader to present it.
Response 8: Although the mechanisms by which truncated O-glycans can contribute to cancer progression are not completely understood, several publications give evidence for some conditions related to hallmark of cancer. Even if an extensive description is out of scope of this review, focusing in therapeutic strategies, we supplement the information as follow: “ These truncated O-glycan antigens are observed at the earliest stages of cellular malignant transformation [127-129] and are significantly associated with tumor progression through various mechanisms affecting adhesion properties of cancer cells, stabilizing receptor expression on the cell surface allowing stronger signaling, and triggering immune suppression by binding to tolerogenic dendritic cells or modulating NK cells action by competitive lectin binding [130-132]. “
Point 9: L427 'built on different backbones such as S*S*S*, S*T*T* or T*T*T* (*=GalNAc). You need to explain what S and T are -- I assume serine and threonine?
Response 9: Thank you for pointing this out. We explain all abbreviations and symbols as follow: (S = Serine; T = Threonine; * = GalNAc).
Point 10: L530 'STn is not expressed by normal cells and tissues' and L596 'TF, like Tn antigen, is found almost exclusively in tumours'. Check whether this is correct. I think Tn antigen is rarely found in normal tissues, commonly in some cancers. TF and STn, I think, are found in relatively low levels in alot of normal tissues, but are present in much higher levels in many cancers. But I don't think it is accurate to say that they are not present in normal. Also avoid using the term 'expressed' when referring to glycans, because, of course, they are not gene products so are not 'expressed', they are synthesised.
Response 10: We agree with the reviewer in that some expressions could induce misinterpretation. In fact, publish results are extremely influenced by used tools. We clarify the text as follow:
“STn is often coexpressed with Tn and, as Tn, its detection is low or null on normal cells or tissues, depending on mAb used.”
“TF is found in relatively low levels in a lot of normal tissues, but is present in much higher level in carcinomas”
We modified the term “expression” whenever possible.
Point 11: L534 'leading to more aggressive cancer behaviour' -- define what is meant by 'aggressive cancer behaviour'. Do you mean, for example, reduced survival?
Response 11: In fact, we allude to pre-clinical evidence for aggressiveness related to STn, and we clarify the information as follow: “De novo STn expression via ST6GalNAc-I transfection can change a tumor’s malignant phenotype [132], leading to more aggressive cancer cell behavior, decreasing cell–cell aggregation and increasing tumor growth, extracellular matrix adhesion, migration, invasion, and metastases [183,184,185].”
Point 12: L636 'human serum albumin' is abbreviated as 'HAS' -- should it be HSA?
Response 12: The reviewer is right; it was certainly a typing error which was corrected.
Point 13: L693 explain what 'FUT enzymes' are
Response 12: We agree with the reviewer that this abbreviation need to be described, and we add “fucosyltransferases (FUTs) enzymes” to the text.
Point 14: Section 5, there are several instances where 'polysialilation' and 'polysialilated' are mis-spelled - they should read polysialylation and polysialylated.
Response 14: Thank you for signaling these typing errors. They were corrected

Reviewer 2 Report
The manuscript “Targeting tumor glycans for cancer therapy: successes, limitations, and perspectives” reviews the current strategies to target tumor glycosylation and their utility for cancer therapy.
The review is well –written and integrates the up to date knowledge of the advances of targeting tumor-associated carbohydrate antigens (TACAs) for cancer therapy.
To improve the review I have some concerns to be addressed:
The authors list the main TACAs described, but not sialyl-Lewis x, which is neo expressed in several carcinomas and considered a TACA. They should include it and comment on it.
In Table 1, they should improve the figures of glycans by indicating the links between monosaccharides and should correct sialyl-Lewis a structure (now it lacks the terminal sialic acid linked alpha-2,3 to a galactose)
Finally, in Figure 1, which represents the antitumor strategies targeting TACAs, they should include/indicate the vaccination strategies that describe along the manuscript.
Author Response
Response to Reviewer 2 Comments
Point 1: The authors list the main TACAs described, but not sialyl-Lewis x, which is neo expressed in several carcinomas and considered a TACA. They should include it and comment on it.
Response 1: We thank the reviewer for pointing this out. We first consider that, as no mAbs or vaccines targeting SLeX have demonstrated usefulness for cancer treatment, it was not relevant for its inclusion in this review. However, considering the reviewer´s recommendation, as well as other aspects of potential relevance in cancer treatment, we add a paragraph of SLeX in the manuscript.
Point 2: In Table 1, they should improve the figures of glycans by indicating the links between monosaccharides and should correct sialyl-Lewis a structure (now it lacks the terminal sialic acid linked alpha-2,3 to a galactose)
Response 2: We agree with the reviewer, SLeA structure was corrected, as well as all linkages were added to the glycan figures.
Point 3: Finally, in Figure 1, which represents the antitumor strategies targeting TACAs, they should include/indicate the vaccination strategies that describe along the manuscript.
Response 3: We modify the Figure 1 as suggested.

Reviewer 3 Report
The review is well written and complete. A large number of studies are considered in this review and although most of them are phase I / II and therefore include agents not yet marketable for readers it represents a good opportunity to broaden one's knowledge in this innovative field.
Author Response
Response to Reviewer 3 Comments
We thank the reviewer for his comment.